# Long-term platinum-based drug accumulation in cancer-associated fibroblasts promotes colorectal cancer progression and resistance to therapy

Jenniffer Linares[1,2], Anna Sallent-Aragay[1], Jordi Badia-Ramentol[1], Alba Recort-Bascuas[1], Ana Méndez[3], Noemí Manero-Rupérez[1], Daniele Lo Re[4], Elisa I. Rivas[1], Marc Guiu[4], Melissa Zwick[1], Mar Iglesias[1,5], Carolina Martinez-Ciarpaglini[6], Noelia Tarazona[5,7], Monica Varese[4], Xavier Hernando-Momblona[4,5], Adrià Cañellas-Socias [4,5], Mayra Orrillo[8], Marta Garrido[1], Nadia Saoudi[9], Elena Elez [9], Pilar Navarro [10,11,12], Josep Tabernero [5,9], Roger R. Gomis [4,5,13], Eduard Batlle [4,5,13], Jorge Pisonero [3], Andres Cervantes [5,7], Clara Montagut[1,5,8] & Alexandre Calon [1] ✉

A substantial proportion of cancer patients do not benefit from platinum-based chemotherapy (CT) due to the emergence of drug resistance. Here, we apply elemental imaging to the mapping of CT biodistribution after therapy in residual colorectal cancer and achieve a comprehensive analysis of the genetic program induced by oxaliplatin-based CT in the tumor microenvironment. We show that oxaliplatin is largely retained by cancer-associated fibroblasts (CAFs) long time after the treatment ceased. We determine that CT accumulation in CAFs intensifies TGF-beta activity, leading to the production of multiple factors enhancing cancer aggressiveness. We establish periostin as a stromal marker of chemotherapeutic activity intrinsically upregulated in consensus molecular subtype 4 (CMS4) tumors and highly expressed before and/or after treatment in patients unresponsive to therapy. Collectively, our study underscores the ability of CT-retaining CAFs to support cancer progression and resistance to treatment.

More than four decades after their first approval, platinum drugs still stand among the most widely utilized anti-cancer agents[1]. Nowadays, almost 50% of cancer patients receiving chemotherapy (CT) are indeed treated with systemic platinum-based regimen[1,2]. Notwithstanding platinum drugs' effectiveness in eradicating cancer cells by means of adducts and crosslinks accumulation in the DNA, a large number of tumors are able to bypass the cytotoxic effect of platinum through primary or acquired resistance[1]. In colorectal cancer (CRC), the improvement in survival provided by oxaliplatin-based regimens is estimated to be no more than 20% in stage III and less than 5% in stage II localized cancer patients[3,4]. Similarly, metastatic CRC patients that are initially responding to therapy often experience disease progression due to the emergence of acquired drug-resistance[5].

For these reasons, elucidating the determinants of platinum-based treatment effectiveness is paramount to improve the standard of care for cancer patients. Much effort has been made to decipher the

mechanisms of platinum resistance intrinsic to epithelial cancer cells[1,6] and multiple molecular classifications were developed in the recent years, enabling the identification of CRC subtypes associated with worse patient's outcome[7–10] and lack of benefit from CT, including oxaliplatin[10,11]. Initially, there was no apparent molecular consensus between the poor prognostic subtypes of CRC, which were either enriched in Wnt signaling and markers associated with cancer stem cells (Stem-like)[7], in serrated tumors (CCS3)[9], or in mesenchymal tumors with deficient mismatch repair machinery (C-type)[10]. In an effort to integrate the previously published classification algorithms, Guinney and colleagues defined four Consensus Molecular Subtypes (CMS) of CRC, namely CMS1 (MSI immune), CMS2 (canonical), CMS3 (metabolic) and CMS4 (mesenchymal), from which CMS4 tumors showed the worse clinical outcome[8]. However, these classification systems and advances in the genetics of cancer have not yet translated into efficient treatment synergizing with platinum drugs and failed so far to provide biomarkers of therapeutic response[12,13].

Importantly, cancer cells do not exist as isolated entities, but rather reside in an interactive tumor microenvironment (TME) composed of non-malignant cells that largely contributes to cancer progression. For instance, the activation of the TGF-beta pathway in cancer-associated fibroblasts (CAFs) populating the TME is considered as a hallmark of worse prognosis for CRC patients whereas low stromal TGF-beta activity associates with increased disease-free survival[14]. In this line, growing evidence indicates that systemic CT affects non-malignant cells in the TME of cancers from distinct origin[15,16]. However, the mechanisms through which CT regulates stromal functions and to what extent these processes influence cancer progression as well as patient's susceptibility to treatment remain unclear.

In this work, we assess the presence of oxaliplatin in residual tumors after treatment and investigate the impact of platinum-based therapy on stroma-originating pro-oncogenic signaling. Surprisingly, we observe that significant amount of oxaliplatin is retained by the TME long time after CRC patient treatment. We functionally dissect the influence of platinum-stimulated TME on cancer progression and demonstrate that the accumulation of platinum in CAFs participates to the acquisition of stromal cues that associate with increased cancer aggressiveness and resistance to treatment.

## Results

### Platinum-based drug accumulates in fibroblasts resilient to treatment

We first aimed to investigate the immediate impact of CT on the TME. For this, we used a model of aggressive CRC grown from mouse tumor organoids (MTO) carrying compound genetic alterations (*Apc, Kras, Trp53, Tgfbr2*) injected into the caecum wall of immunocompetent *C57BL/6J* mice (see "Methods")[17]. Mice were administered with oxaliplatin 96 and 24 h before tumor resection (see "Methods"). Bioluminescence tracking in vivo showed a reduction of cancer cells abundance upon treatment (Fig. 1a). Following resection, tumor samples were analyzed by immunohistochemistry (IHC) to assess α-SMA (+) CAFs, CD31 (+) endothelial and CD45 (+) immune cells abundance. We observed a significant reduction of blood vessels density and immune cells infiltration following therapy (Fig. 1a and Supplementary Fig. 1a). On the other hand, CAFs abundance remained unchanged (Fig. 1a and Supplementary Fig. 1a) therefore suggesting that in contrast to cancer, endothelial and immune cells, CAFs are highly resistant to oxaliplatin. We next assessed oxaliplatin cytotoxicity against cultured colonic fibroblasts (CCD-18Co), CAFs derived from CRC patients (CAF1, CAF2, CAF3), immune cells (PBMC), endothelial cells (HUVEC), CRC cell line (HT29-M6) and patient-derived CRC organoids (PDO, PDO2). In accordance with in vivo findings, we observed that CRC (Fig. 1b and Supplementary Fig. 1b), immune (Supplementary Fig. 1c) and endothelial cells (Supplementary Fig. 1d) were highly sensitive to oxaliplatin compared to fibroblasts (Fig. 1b and Supplementary Fig. 1e).

In order to determine the tumor cells response to CT over time, we treated cultured CCD-18Co, CAF1 and CAF2 as well as HT29-M6 cells, PDO and PDO2 with oxaliplatin for 12 days. Cancer cells did not survive to 9 days of treatment (Fig. 1c and Supplementary Fig. 1f). In contrast, about 50–80% of fibroblasts resisted to up to 12 days CT (Fig. 1c and Supplementary Fig. 1f). Yet, ICP-MS analysis indicated an increased oxaliplatin absorption in cultured fibroblasts compared to CRC cells (Fig. 1d). Interestingly, traces of platinum were still detectable in fibroblasts long after oxaliplatin retrieval (Fig. 1d).

Prompted by these findings, we sought explore the patterns of platinum drug biodistribution in tumors from CRC patients. To this end, we applied laser ablation inductively coupled plasma mass spectrometry (LA-ICP-MS) to the detection of oxaliplatin in samples from patients with advanced CRC. Tumor tissues were obtained from surgical resection achieved few days and up to 2.5 years after CT (see "Methods"). Histological analysis by expert pathologists and LA-ICP-MS bioimaging on formalin-fixed paraffin-embedded (FFPE) tumor sections stained with hematoxylin and eosin revealed that oxaliplatin was predominantly retained in tumor areas enriched with fibroblasts (Fig. 1e, f and Supplementary Fig. 2a). Accordingly, IHC analysis indicated that the expression of FAP—a marker of CAFs—overlapped with tumor areas displaying increased platinum uptake (Supplementary Fig. 2b). Of note, oxaliplatin uptake remained measurable at least 845 days after the last cycle of therapy (Fig. 1f and Supplementary Fig. 1a).

Collectively, our data argue that in contrast to any other cell subtype present in the tumor, fibroblasts—the most prominent cell type within the TME[18,19]—display great resistance to oxaliplatin cytotoxicity and are particularly prone to absorb and retain platinum-based drug. These results also suggest that endothelial and immune cells present at the time of CT are cleared from the tumor, being eventually replaced over time by cells recruited after the treatment ceased. Therefore, our findings indicate that long-term platinum retention after CT occurs in the TME of CRC patients and suggest that this process is largely dependent on resilient CAFs present at the time of treatment.

### Platinum-stimulated fibroblasts promote CRC progression

We sought to functionally dissect the effect of oxaliplatin-retaining fibroblasts on cancer progression. To this end, HT29-M6 cells were inoculated into nude mice together with CCD-18Co that had been previously treated in vitro with oxaliplatin. Cells were transplanted subcutaneously in quantities that generated suboptimal engraftment in control conditions. In contrast to the non-treated ones, oxaliplatin-preactivated fibroblasts significantly reduced disease latency and increased the engraftment of cancer cells in recipient mice (Fig. 2a). After initiation, tumor growth followed a similar trend in both conditions (Supplementary Fig. 3a), suggesting that pre-treated fibroblasts co-injected with CRC cells promote tumor initiation rather than tumor expansion. This result may also be explained by the transitory impact of co-injected fibroblasts on the tumor development. Indeed, mouse stroma is actively recruited during tumor expansion and CAFs of murine origin are replacing co-injected fibroblasts over time[14]. Next, we administrated the conditioned media (CM; see "Methods") obtained from oxaliplatin-stimulated CCD-18Co to cultured HT29-M6 cells that were then treated with oxaliplatin. We realized that pre-treated CCD-18Co CM significantly enhanced cancer cells resistance to platinum-based therapy (Fig. 2b). Conversely, CM derived from pre-treated cancer, endothelial or immune cells failed to protect target CRC cells from CT (Supplementary Fig. 3b).

These findings led us to investigate whether the gene program induced by platinum absorption in fibroblasts may be enriched with markers inherently associated with disease aggressiveness in CRC patients. For this, we used as surrogate the set of genes identified by

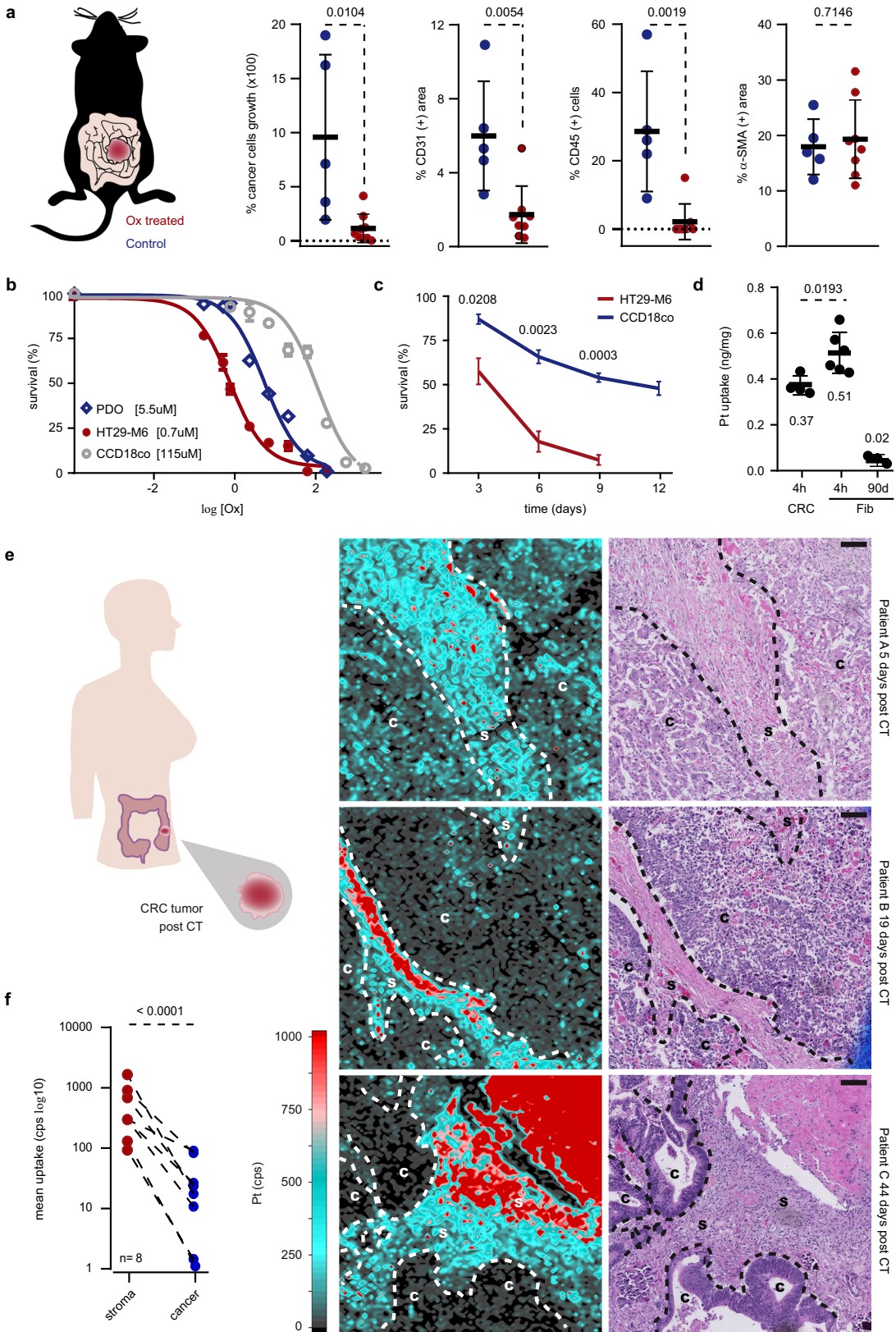

transcriptomic analysis in CCD-18Co activated by oxaliplatin (aFib-RS; activated fibroblasts response signature; see "Methods" and Supplementary Data 1). We applied gene set enrichment analysis (GSEA) to the study of aFib-RS in CRC cells and CAFs purified from patient tumors (GSE39396)[14]. GSEA indicated that aFib-RS was enriched in CAFs compared to cancer cells, thus suggesting that genes in the aFib-

RS are preferentially expressed by CAFs (Supplementary Fig. 3c). In order to decouple the confounding effect of the CAFs hallmarks, we identified the CAF-specific and CAF non-specific aFib-RS probe sets (FAP (+) aFib-RS and FAP (−) aFib-RS respectively) in FAP (+) CAFs compared to CD31 (+), CD45 (+) or EpCAM (+) cell subtypes (GSE39396; FC > 1; $p < 0.05$; FDR < 0.01; Supplementary Data 1).

**Fig. 1 | Platinum accumulates within fibroblasts resilient to treatment. a** Tumor analysis in MTO-injected *C57BL/6J* mice treated with oxaliplatin. Left panel: percentage of tumor growth upon treatment. Right panels: percentage of CD31, CD45 and α-SMA intratumoral positivity. Control (blue; *n* = 5) and treated conditions (red; *n* = 8). Values are mean ± sd. *p* values are indicated. Drawing modified from Tauriello et al.[17]. **b** Biological activity of oxaliplatin against HT29-M6, PDO and CCD-18Co. Values are mean ± sd. EC$_{50}$ are indicated. Representative of *n* = 3 biologically independent experiments. **c** 12-days follow-up of HT29-M6 and CCD-18Co cells survival upon oxaliplatin treatment. *n* = 3 biologically independent experiments. Values are mean ± sd. *p* value is indicated. **d** Oxaliplatin uptake in HT29-M6 (CRC) after 4 h treatment and in CCD-18Co (Fib) after 4 h treatment and 90 days after treatment. *n* = 6 biologically independent experiments. Mean quantities ± sd and *p* value are indicated. **e** Oxaliplatin biodistribution representative of eight independent patient tumors resected after CT. Time post CT is indicated. Left panels: platinum (Pt) uptake map. Right panels: corresponding hematoxylin and eosin staining, Scale bars: 100 μm. Drawing modified from Rivas, Linares et al.[78]. **f** Mean stromal (red) and cancer (blue) Pt uptake in samples from eight CRC patients. *p* value is indicated. PDO patient-derived tumor organoids, C cancer, S stroma, CT chemotherapy, cps count per second. Two-sided, unpaired (**a, c, d**) and paired (**f**) *t*-test *p* values (*p*) are indicated. Source data are provided as a Source Data file.

We tested the predictive value of each probe set in two representative cohorts of 177 (GSE17536) and 519 CRCs (GSE39582) for which transcriptomic profiles and clinical history were publicly available[20,21]. High aFib-RS was an independent predictor of worse outcome in both cohorts (Fig. 2c, d and Supplementary Table 1a). Similar results were obtained with FAP (+) and FAP (−) aFib-RS (Supplementary Table 1b). We realized that aFib-RS levels were distinctly upregulated in CMS4 tumors (Fig. 2e), a CRC subtype characterized by abundant stroma[8]. In order to test their prognosis value in tumors with high stromal content, we further analyzed our signatures in the CMS4 population from GSE39582 (*n* = 121; see "Methods"). Remarkably, high expression of the full aFIB-RS but also of FAP (+) and FAP (−) aFib-RS subsets robustly predicted poor outcome in this particular subtype (Fig. 2f and Supplementary Table 1b), thus indicating the superior predictive power of aFib-RS over CMS.

We wondered whether aFib-RS levels may identify as well patients unresponsive to treatment. To address this question, we selected transcriptomic data from two subsets of advanced CRC patients that had received adjuvant platinum-based CT (FOLFOX) after intended curative surgery either annotated for response to therapy status (GSE72970; 32 stage III–IV patients) or disease relapse (GSE14333; 51 stage III patients)[22,23]. GSEA showed that the levels of aFib-RS, FAP (+) aFib-RS and FAP (−) aFib-RS were significantly upregulated in tumors from patients unresponsive (Fig. 2g and Supplementary Table 1c) or relapsing after CT (Fig. 2h and Supplementary Table 1c). While not reaching statistical significance, an upregulation trend of aFib-RS did also associate with resistance to CT in the limited (*n* = 10) CMS4 population identified in GSE14333 (Supplementary Fig. 3d). Altogether, these data indicate an association between the gene program activated in fibroblasts upon oxaliplatin treatment and the lack of benefit from CT.

### Stromal platinum accumulation induces an autocrine activation of the TGF-beta pathway

We next sought to achieve a comprehensive analysis of the molecular pathways activated by oxaliplatin in resilient fibroblasts. To do so, we used as surrogates ~220 gene sets representing biological states, processes and cellular pathways dysregulated in cancer, either available publically or generated by manual curation[24]. We determined by GSEA that hallmarks of P53 pathway, DNA repair and cellular apoptosis were among the top signatures enriched upon treatment in cultured fibroblasts, therefore underscoring the cytotoxic stress imposed by oxaliplatin to these cells (Supplementary Table 2a). However, GSEA also revealed a robust upregulation of genes associated with senescence (SenRS; Senescence response signature) and senescence-associated secretory phenotype (SASP-S; SASP signature) (Fig. 3a). Corroborating these data, fibroblasts displayed high mRNA levels of senescence and SASP markers−*CDKN1A*[25], *CDKN2A*[25], *IL1B*[26], *CXCL8*[26], *GDF15*[27] (Fig. 3b and Supplementary Fig. 4a) as well as increased senescence-associated β-galactosidase activity following treatment with oxaliplatin (Fig. 3c and Supplementary Fig. 4b). Of note, we observed that CAFs are intrinsically enriched in SASP-S

compared to CRC cells (Supplementary Table 2b). Yet, we realized that while given senescence markers are more abundantly expressed by CAFs (CXCL8, CDKN1A), others (IL1B, GDF15, CDKN2A) are either enriched in CRC cells or equally expressed by both cell subtypes (Supplementary Fig. 4c). As for aFib-RS, we identified FAP (+) and FAP (−) SASP-S probe sets (Supplementary Data 1) and tested their capacity to segregate relapsing and non-relapsing patients as well as patients responsive and unresponsive to CT. FAP (+) SASP-S failed to identify disease recurrence in the CMS4 subset (Supplementary Table 2c) and relapsing patients after therapy (Supplementary Table 2d). Yet, SASP-S and FAP (−) SASP-S were upregulated in relapsing and unresponsive patients in all cases (Fig. 3d and Supplementary Table 2c, d).

Given the above findings, we hypothesized that the pro-tumorigenic support provided by platinum-stimulated fibroblasts may depend on secreted factors related to SASP. In this line, the analysis of paired tumor tissue samples obtained from CRC patients before and after CT (see "Methods") revealed that the mRNA expression of TGF-beta 1, a SASP factor[28–30] associated with stromal activation and progression to metastasis in CRC[14], was significantly increased after CT (Fig. 3e). We observed that *TGFB1* upregulation over time was also detected in cultured fibroblasts following oxaliplatin treatment (Fig. 3f and Supplementary Fig. 4d). In contrast, this increase was not apparent in any other tumor cell subtype upon treatment (Supplementary Fig. 4e). The levels of both *TGFB1* and the gene expression program activated by TGF-beta 1 in colon fibroblasts (Fib-TBRS; see "Methods") were robustly correlated with aFib-RS in patients' tumors (Fig. 3g), therefore indicating that these processes are concurrently modulated in CRC. Correspondingly, the Fib-TBRS was significantly enriched in oxaliplatin-treated CCD-18Co (Fig. 3h). We further explored TGF-beta pathway activation by IHC in MTO-injected mice using P-SMAD3 as a surrogate marker of TGF-beta activity[14]. Of note and similar to a significant proportion of aggressive CRCs, MTOs are unresponsive to TGF-beta due to compound mutation in the TGF-beta receptor 2 (Tgfbr2). We observed that P-SMAD3 was increased upon oxaliplatin treatment in the TME of MTO-injected mice (Supplementary Fig. 4f, g). Corroborating this data, CCD-18Co displayed increased levels of P-SMAD3 upon oxaliplatin administration (Supplementary Fig. 4h). We next used Fib-TBRS levels as an indicator of TGF-beta activity in patients and explored its association with clinical outcome. In contrast to aFib-RS, high Fib-TBRS expression failed to segregate relapsing patients in the CMS4 CRC subtype (Supplementary Table 2c), yet elevated expression of Fib-TBRS and FAP (+) or (−) Fib-TBRS subsets were associated with an increased risk of relapse in the overall CRC population (Supplementary Table 2c). With the exception of the FAP (+) subset in GSE72970, Fib-TBRS and FAP (+) or (−) Fib-TBRS were also dramatically upregulated in tumors unresponsive to CT (Fig. 3i and Supplementary Table 2d).

Collectively, these data indicate that the cellular stress imposed by long-lasting accumulation of platinum to CRC-associated fibroblasts may influence disease progression and response to therapy by increasing TGF-beta activity in the TME.

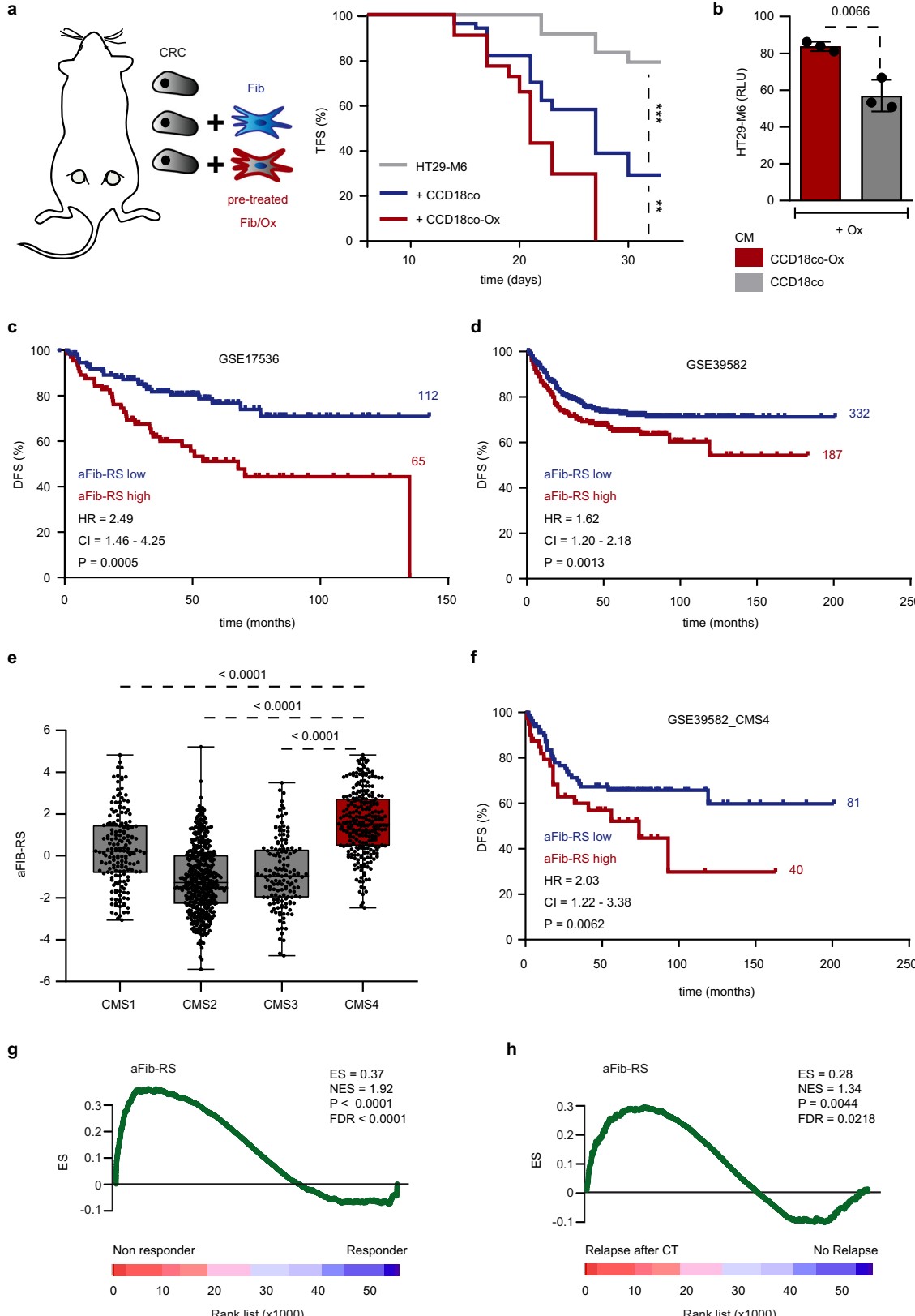

## Platinum drug induces a pro-oncogenic secretory phenotype in CRC fibroblasts

IHC analyses in tumors from MTOs-injected mice showed increased P-STAT3 nuclear staining after therapy (Supplementary Fig. 5a), indicating the activation of STAT3 signaling upon treatment. Remarkably, the expression of interleukin 11 (IL11), one of the activators of STAT3

phosphorylation[31] previously reported as a downstream effector of the TGF-beta pathway involved in cancer progression[14], was significantly correlated with aFib-RS levels in CRC patients (Fig. 4a). In line with this, IL11 expression was upregulated in CCD-18Co (Fig. 4b and Supplementary Fig. 5b), CAF1 and CAF2 (Fig. 4c) upon oxaliplatin administration. *IL11* upregulation was maintained over time after treatment (Fig. 4d) and

**Fig. 2 | Platinum-stimulated fibroblasts promote CRC progression.**
**a** Kaplan–Meier plot displays tumor initiation overtime in nude mice injected subcutaneously with 15,000 HT29-M6 cells alone (gray; $n = 24$), co-injected with 50,000 CCD-18Co non-treated (blue; $n = 50$) or pre-treated with oxaliplatin (red; $n = 44$). **$p = 0.0015$, ***$p = 0.0002$. Drawing modified from Calon et al.[33].
**b** Quantitative analysis of oxaliplatin-treated HT29-M6 cells cultured with conditioned media (CM) from CCD-18Co non-treated or pre-treated with oxaliplatin. $n = 3$ biologically independent experiments. Values are mean ± sd. $p$ value is indicated. **c** Kaplan–Meier curve displays DFS for GSE17536 patients ($n = 177$) presenting low (blue; $n = 112$) or high expression levels of aFib-RS (red; $n = 65$). HR, CI, $p$ value are indicated. **d** Kaplan–Meier curve displays DFS for GSE39582 patients ($n = 519$) presenting low (blue; $n = 332$) or high expression levels of aFib-RS (red; $n = 187$). HR, CI, $p$ value are indicated. **e** aFib-RS levels in $n = 1029$ CRC patients classified by CMS subtypes (CMS1 $n = 175$; CMS2 $n = 445$; CMS3 $n = 147$, CMS4 $n = 262$). Central mark indicates the median, box extends from the 25th to 75th percentiles, whiskers represent the maximum and minimum data point. $p$ values are indicated. **f** Kaplan–Meier curve displays DFS for GSE39582_CMS4 patients ($n = 121$) presenting low (blue; $n = 81$) or high expression levels of aFib-RS (red; $n = 40$). HR, CI, $p$ value are indicated. **g** GSEA of aFib-RS in the GSE72970 subset of tumor samples collected before treatment comparing patients responding to CT ($n = 20$) and unresponsive ($n = 12$) patients. **h** GSEA of aFib-RS in the GSE14333 subset of tumor samples collected before treatment comparing relapsing ($n = 13$) to non-relapsing ($n = 38$) patients after CT. ES enrichment score, NES normalized enrichment score, FDR false discovery rate. TFS tumor-free survival, CRC colorectal cell line (HT29-M6). Fib fibroblasts (CCD-18Co), DFS disease-free survival, HR hazard ratio, CI confidence interval, Ox oxaliplatin, GSEA gene set enrichment analysis, RLU relative luminescence unit. Two-sided, unpaired $t$-test $p$ values ($p$) are indicated for **b**, **e**. Log-rank (Mantel–Cox test) $p$ values ($p$) are indicated for **a**, **c**, **d**, **f**. GSEA nominal $p$ value ($p$) and FDR-adjusted $p$ value are indicated for **g**, **h**. Source data are provided as a Source Data file.

reduced upon administration of TGF-beta pathway inhibitor (Fig. 4e). In addition and similar to aFib-RS, high *IL11* mRNA expression was an independent predictor of poor prognosis in the two analyzed patient cohorts (Fig. 4f and Supplementary Fig. 5c, d). Accordingly, *IL11* levels were distinctly upregulated in CMS4 tumors (Fig. 4g). As for aFib-RS, the increased expression of *IL11* robustly segregated relapsing patients in this particular subset (Fig. 4h and Supplementary Fig. 5d).

In order to assess the relevance of IL11 pathway activation in patients treated with CT, we used as surrogate a signature of response to IL11 in CRC cells (CRC-IL11RS)[14]. We observed that CRC-IL11RS was robustly upregulated in patients either unresponsive to therapy (Supplementary Fig. 5e) or relapsing after treatment (Fig. 4i). We thus investigated whether increased intratumoral levels of IL11 may enhance tumor initiation in CRC. To this end, we inoculated nude mice subcutaneously with epithelial CRC cells (HT29-M6) engineered to autonomously produce IL11 (CRC-IL11). Forced secretion of IL11 drastically increased the frequency of tumor formation and reduced the disease latency period (Fig. 4j).

We next explored the relation between stromal TGF-beta pathway activation and IL11 activity in epithelial CRC cells during cancer progression. To this end, we engineered HT29-M6 cells simultaneously knocked down for IL11 receptor subunit alpha (IL11RA) and secreting active TGF-beta 1 that we injected subcutaneously into nude mice. The elevated secretion of TGF-beta 1 by shCt/TGF-beta 1 cancer cells (CRC-shCt/T) robustly enhanced tumor initiation compared to control (CRC-shCt) and to CRC-shIL11RA cells (Fig. 4k; dashed red, dashed blue and red lines). However, reducing the expression of IL11RA restricted the capacity of shIL11RA/TGF-beta 1 CRC cells (CRC-shIL11RA/T) to initiate tumors in a TGF-beta enriched environment (Fig. 4k; blue line).

Taken together, these data indicate that platinum uptake induces TGF-beta pathway autocrine activation in stromal cells leading to the production of IL11. In turn, secreted IL11 is capable of enhancing epithelial CRC cells tumor initiation capacity.

## POSTN is a marker of oxaliplatin-induced TGF-beta activity in the TME

Given the robust association of aFib-RS whole gene program with poor outcome and resistance to therapy, we hypothesized that additional stromal biomarkers of response to CT may be critical for CRC patient treatment. We thus extended our analysis of genes modulated by oxaliplatin in fibroblasts and identified periostin (POSTN), a CAF-specific secreted factor[32–34] promoting CRC progression and metastasis[35,36], as one of the most upregulated gene in aFib-RS (Fig. 5a and Supplementary Fig. 6a, b). IHC evaluation in consecutive sections of CRC tumors performed by expert pathologist indicated that POSTN and FAP were expressed by tumor-associated fibroblasts (Supplementary Fig. 6c), thus corroborating the POSTN expression by CRC CAFs reported in previous studies[32,33]. We observed that POSTN levels were largely upregulated in CCD-18Co, CAF1 and CAF2 upon oxaliplatin administration (Fig. 5b, c and Supplementary Fig. 6d). Accordingly, IHC analysis in tumors from MTOs-injected mice showed increased stromal expression of POSTN following oxaliplatin administration (Fig. 5d). Oxaliplatin-induced *POSTN* expression was maintained after treatment over time (Supplementary Fig. 6e) and was abrogated upon administration of a TGF-beta inhibitor (Fig. 5b). Accordingly, *POSTN* expression correlated robustly with aFib-RS, Fib-TBRS and *TGFB1* in CRC patients (Fig. 5e).

These findings led us to test the clinical utility of POSTN protein expression as a stromal marker of chemotherapeutic and/or TGF-beta activity in FFPE samples from CRC patients. It is worth mentioning that CRC patients are currently treated with oxaliplatin combination regimens rather than oxaliplatin alone, thus complicating the determination of platinum specific impact on POSTN expression. To overcome this problem, we used an experimental approach allowing the ex vivo maintenance and treatment of tumor explants that recapitulate the intratumoral heterogeneity—CRC cells and TME—specific to a given patient (see "Methods"). For this, freshly resected tumor samples from four untreated CRC patients were administered ex vivo with oxaliplatin monotherapy. IHC analysis indicated that treated tumors displayed abundant stromal accumulation of POSTN compared to untreated tumor samples from the same patients (Fig. 5f, g; upper and middle panels). The addition of a TGF-beta pathway inhibitor to the regimen largely reduced POSTN upregulation ex vivo (Fig. 5f, g; lower panel), therefore indicating that POSTN protein expression associates with TGF-beta pathway autocrine activation occurring upon oxaliplatin administration.

## POSTN protein levels identify CMS4 subtype and tumors unresponsive to therapy

Similar to aFib-RS, POSTN mRNA levels were increased in the CMS4 patient subset (Supplementary Fig. 7a) and independently predicted poor outcome in the overall CRC population (Supplementary Fig. 7b–d). POSTN mRNA segregated as well relapsing and non-relapsing patients in the CMS4 subtype (Supplementary Fig. 7d, e).

In view of the clinical need for biomarkers that can be evaluated by IHC, we assessed the prognostic value of POSTN protein expression patterns in FFPE tumor samples from a cohort of 109 CMS-classified CRC patients (IHC cohort)[37]. Staining analyses showed that stromal POSTN expression (Fig. 6a) and stromal abundance (Fig. 6b) were both significantly increased in CMS4 tumors compared to any other CMS. However, the tumor stromal load did not correlate with POSTN staining intensity (Supplementary Fig. 7f). Accordingly, only elevated POSTN protein expression predicted decreased disease-free survival (DFS) (Fig. 6c and Supplementary Fig. 7d) while stromal abundance failed to segregate patients with higher risk of relapse (Fig. 6d). We reproduced POSTN protein analyses in the subset of 39 CMS4 cases from IHC cohort. Here again, POSTN expression did not correlate with

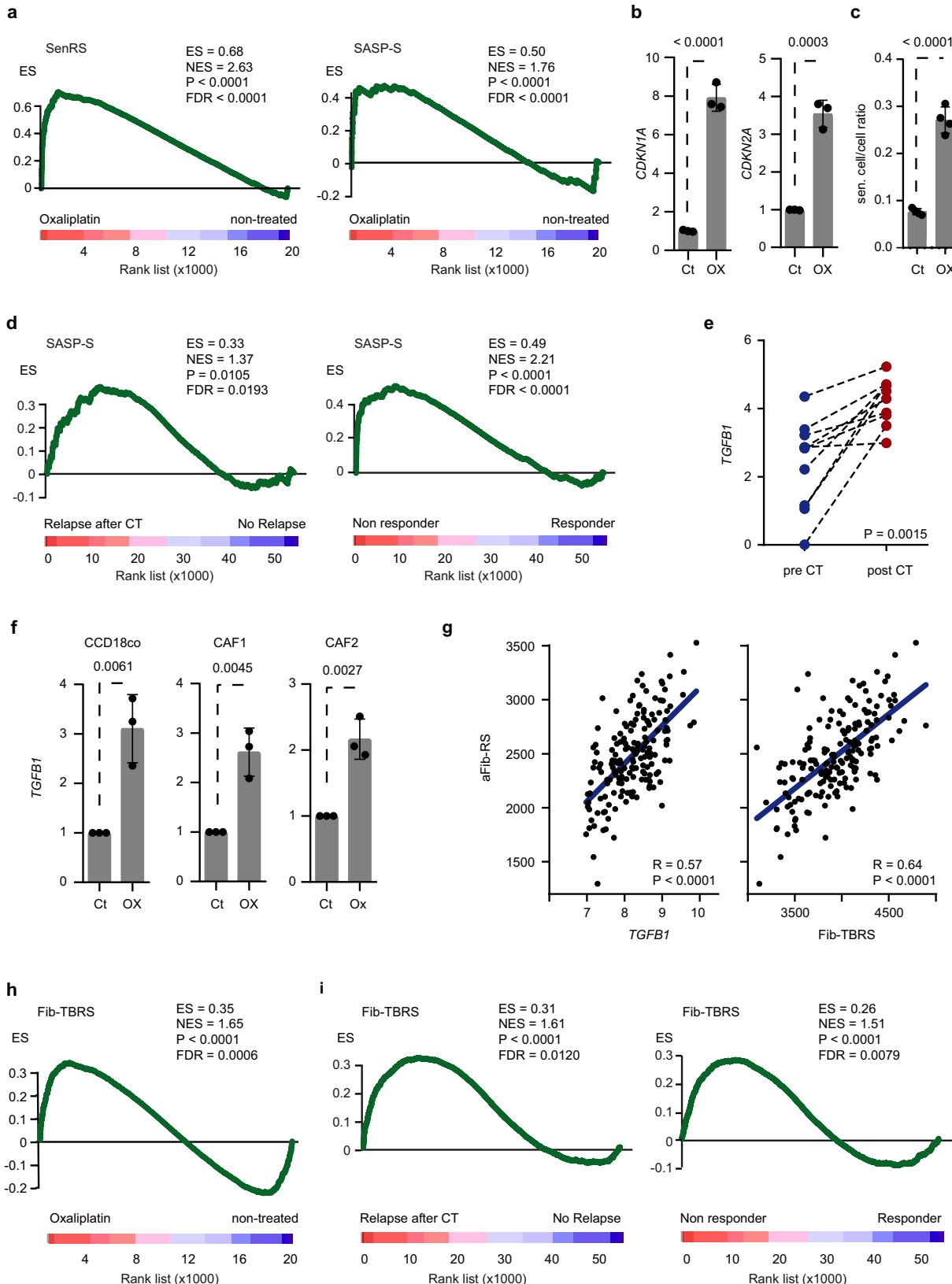

stromal load (Supplementary Fig. 7g) and increased stromal abundance did not associate with higher risk of relapse (Supplementary Fig. 7h). Yet, low POSTN intensity segregated a small subset of CMS4 patients with no observed recurrences (Supplementary Fig. 7i). Although it did not reach statistical significance, this result corroborates the transcriptomic data that associate increased *POSTN* levels

with reduced survival in CMS4 patients (Supplementary Fig. 7d,e). Prompted by these findings, we evaluated the prognosis power of POSTN protein expression in tumors with low or high stromal content. Remarkably, increased POSTN levels was associated with worse outcome in patients with tumors displaying either high (>30%; Fig. 6e) or low (<30%; Fig. 6f) stromal load. Collectively, these data indicate that

**Fig. 3 | Platinum absorption increases TGF-beta activity in fibroblasts. a** GSEA of SenRS (left panel) and SASP-S (right panel) comparing oxaliplatin treated (n = 2) and non-treated (n = 2) CCD-18Co in culture. **b** Relative expression levels of *CDKN1A* and *CDKN2A* in CCD-18Co treated with oxaliplatin. n = 3 biologically independent experiments. Values are mean ± sd. p values are indicated. **c** β-galactosidase positive CCD-18Co quantification after oxaliplatin treatment. n = 4 biologically independent experiments. Values are mean ± sd. p value is indicated. **d** GSEA of SASP-S comparing relapsing (n = 13) to non-relapsing (n = 38) patients after CT (GSE14333; left panel) and comparing patients responding to CT (n = 20) to unresponsive (n = 12) patients (GSE72970; right panel). **e** Relative expression levels of *TGFB1* in paired tumor samples from ten CRC patients before and after platinum-based CT. p value is indicated. **f** Relative expression levels of *TGFB1* in CCD-18Co, CAF1 and CAF2 treated with oxaliplatin. n = 3 biologically independent experiments. Values are mean ± sd. p values are indicated. **g** Correlation between aFib-RS and Fib-TBRS or *TGFB1* in n = 177 CRC tumors from GSE17536. Correlation values (R) and Spearman p values are indicated. **h** GSEA of Fib-TBRS comparing oxaliplatin treated (n = 2) and non-treated (n = 2) fibroblasts. **i** GSEA of Fib-TBRS comparing relapsing (n = 13) to non-relapsing (n = 38) patients after CT (GSE14333; left panel) and comparing patients responding to CT (n = 20) to unresponsive (n = 12) patients (GSE72970; right panel) GSEA gene set enrichment analysis, ES enrichment score, NES normalized enrichment score, FDR false discovery rate, CT chemotherapy, Ox oxaliplatin, Ct control. Two-sided, unpaired (**b**, **c**, **f**) and paired (**e**) t-test p values (p) are indicated. GSEA nominal p value (p) and FDR-adjusted p value are indicated for **a**, **d**, **h**, **i**. Source data are provided as a Source Data file.

POSTN is a robust prognostic marker of disease relapse, reflecting stromal activation state rather than stromal abundance in CRC including CMS4 tumors.

As mentioned previously, recent studies have associated the CMS4 subtype with a reduced susceptibility to systemic therapies[8,11,38]. We thus evaluated the clinical potential of POSTN protein expression as a stromal marker of response to CT. We performed IHC analyses in tumors from 28 advanced CRC patients obtained before and after CT and realized that POSTN levels were overall upregulated after treatment (Fig. 6g; gray box). Remarkably, a comparative analysis of patients responsive (R, n = 17) or unresponsive to treatment (NR, n = 11) tumors showed that POSTN expression was already significantly higher in unresponsive tumors samples obtained before treatment compared to the responsive ones (Fig. 6g; blue dots). This difference was further accentuated between R and NR tumor samples obtained after CT (Fig. 6g; red dots). Analysis of pre and post CT paired samples indicated that NR tumors were either inherently displaying high levels of POSTN before CT or increased their expression of POSTN upon CT (Fig. 6g, h). In contrast, POSTN levels were low before CT and remained unchanged after CT in patients responding to therapy (Fig. 6g).

### High expression of POSTN isoform 4 reduces CRC susceptibility to treatment

We next investigated the contribution of POSTN to the reduced susceptibility to CT observed in resistant CRC tumors. Of note, eight POSTN splicing variants with distinct patterns of expression have been reported in human[39]. A comparative analysis revealed that POSTN isoform 4 (POSTNi4; NM_001135936.2) was the variant most upregulated in CCD-18Co upon CT (Fig. 7a). Similarly, oxaliplatin robustly increased *POSTNi4* levels in CAF1 and CAF2 (Fig. 7b). Furthermore, *POSTNi4* expression was maintained over time after treatment (Fig. 7d) and TGF-beta pathway inhibition abrogated *POSTNi4* upregulation (Fig. 7c).

Given the prevalence of this variant, we sought to assess the contribution of POSTNi4 to cancer progression. We engineered patient-derived organoids (PDO) to autonomously produce this secreted factor (PDO-POSTNi4; Fig. 7e) and inoculated them subcutaneously into NSG mice. Forced POSTNi4 expression didn't impact either tumor initiation (Supplementary Fig. 8a) or tumor expansion (Supplementary Fig. 8b). We next treated macroscopic tumor-bearing mice with oxaliplatin for 20 days and monitored tumor growth kinetics upon therapy. Oxaliplatin treatment significantly reduced tumors expansion in control conditions (Fig. 7f; left panel). In contrast, the growth rate of tumors enriched with POSTNi4 were not affected by therapy (Fig. 7f; right panel), thus suggesting that elevated levels of POSTNi4 increase the resistance to CT in CRC tumors.

## Discussion

Current CRC standard-of-care involves either fluoropyrimidines (5-fluorouracil, capecitabine) alone or in combination with oxaliplatin. As demonstrated by several studies, oxaliplatin administration offers little additive benefit to fluoropyrimidines, and a high proportion of patients will still relapse/progress[1,2,40]. Yet, the addition of oxaliplatin to fluoropyrimidines regimen remains one of the main treatment option in CRC[40–42]. Hence, much efforts are being made to decipher and improve the response to oxaliplatin in epithelial CRC cells[5]. Conversely, data reporting the impact of oxaliplatin on non-malignant cells from the TME are still scarce[15,16].

Here, we have characterized the pattern of oxaliplatin accumulation in tumor tissue from CRC patients using elemental imaging technique originally developed in the field of geology and adapted to biological samples[43]. Surprisingly, we realized that the TME is the main tumor compartment retaining oxaliplatin long after the treatment ceased. We hypothesized that stroma reprogramming by platinum may diminish the drug cytotoxicity against epithelial cancer cells and impact cancer progression. In agreement with this assumption, our data argue that oxaliplatin incorporation induces a senescent state in resilient fibroblasts, which leads to enhanced SASP supporting residual cancer cells survival and aggressiveness. As previously reported by multiple works, TGF-beta is one of the components of the SASP[44–46] and is mainly expressed by CAFs in CRC[47–49]. Our results indicate that oxaliplatin-induced SASP in CAFs leads to increased TGF-beta production, which in turn activates the downstream TGF-beta pathway[14,49,50]. Accordingly, both aFib-RS and Fib-TBRS described in this study could be used to determine tumors with poor outcome according to their stromal composition. However, the predictive power of aFib-RS remains superior to that of Fib-TBRS in stromal (CMS4) CRC tumors, which suggests that expression of aFib-RS can further identify CRC with tumor promoting/protective stroma.

Resulting from the association between platinum absorption and the activation of the TGF-beta pathway, CAFs upregulate IL11 and POSTNi4 expression upon oxaliplatin treatment. Our data show that, in turn, increased IL11 and POSTNi4 respectively contribute to enhanced tumor initiation and resistance to therapy, thus providing original insights on the mechanism behind relapses in CRC patients that fail to respond to platinum-based CT. Although in non-treatment conditions, the expression of both IL11 and POSTN could be mediated by other signaling pathways such as oncogenic Ras for IL11[51] or IL6 and Notch for POSTN[35,52], TGF-beta remains the main inducer of IL11 and POSTN expression in CAFs[14,33,34,53–58]. Of note, *TGFB1*, *IL11*, *POSTN* and *POSTNi4* upregulation remained detectable in cultured fibroblasts long after the treatment ended. Yet, their expression consistently followed a decreasing trend over time paralleling the one observed for oxaliplatin uptake, thus suggesting a potential association between platinum intracellular concentration and TGF-beta activity.

The fact that platinum-activated TME may weaken CT efficacy does not undermine the direct cytotoxicity of platinum-based CT against epithelial cancer cells. Yet, our findings advocate for combinatorial therapies involving senolytics[59,60], TGF-beta pathway inhibitors[61] or anti-IL11 therapies[62] to improve the response to current platinum-based regimens. Our data further suggest that future

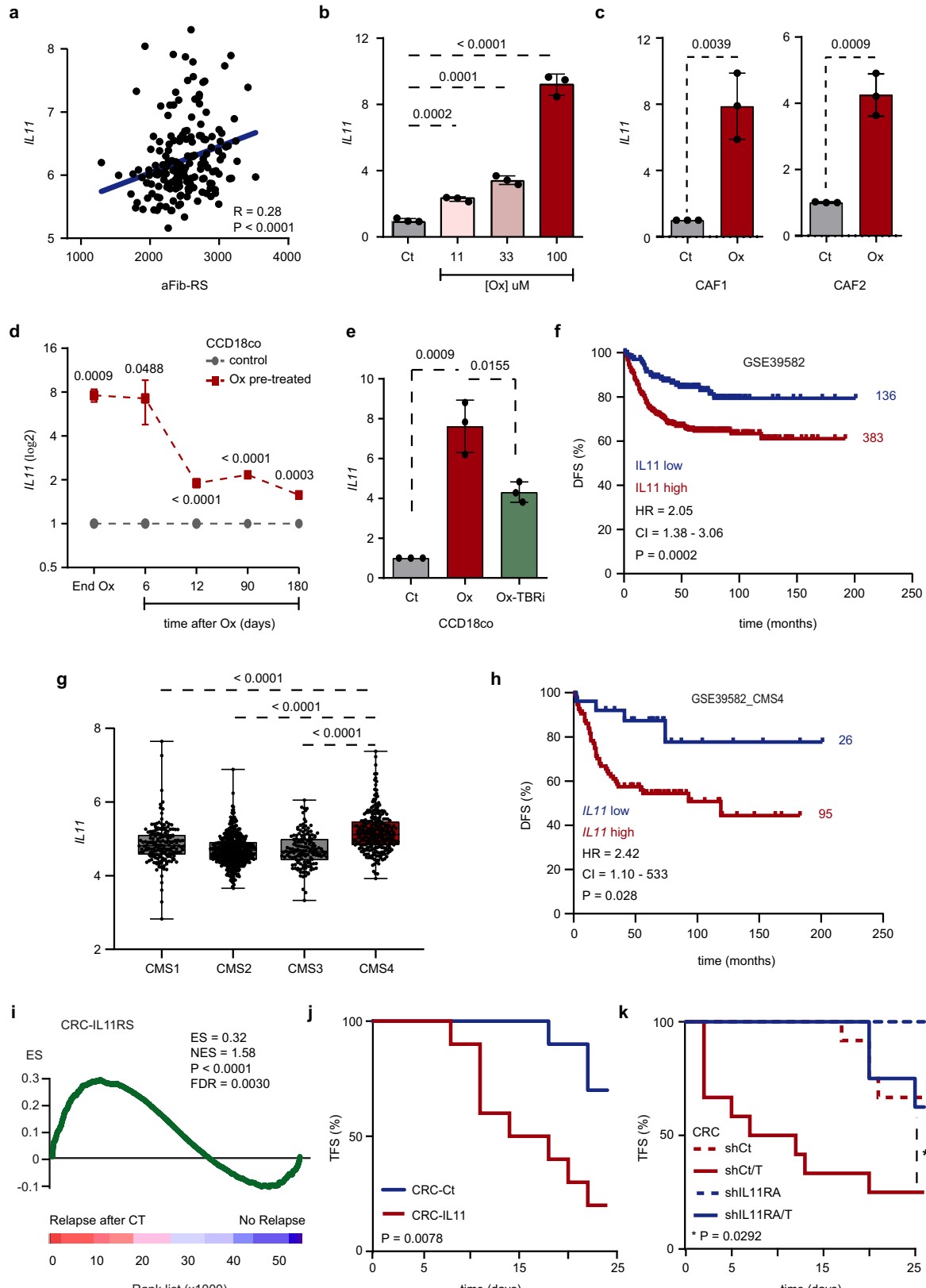

strategies blocking POSTNi4 expression may restore treatment efficiency in unresponsive patients. Transcriptomic analyses indicate that multiple additional soluble factors are upregulated in CAFs stimulated by platinum-based therapy (Supplementary Data 1). Among them, VCAN[63], HBEGF[64], NRG1[65,66], GDF15[27], MMP1[67] and ANGPTL4[68]

were also associated with increased cancer aggressiveness. These findings suggest a coordinate activation of pro-oncogenic signals originating from non-malignant stromal cells upon platinum absorption and provide an extensive array of potentially actionable biomarkers in CRC.

**Fig. 4 | TGF-beta pathway autocrine activation in platinum-stimulated fibroblasts upregulates IL11 secretion. a** Correlation between *IL11* and aFib-RS in *n* = 177 CRC tumors from GSE17536. Correlation (*R*) and Spearman *p* value are indicated. **b** Relative expression levels of *IL11* in CCD-18Co treated with increasing concentration of oxaliplatin. *n* = 3 biologically independent experiments. Values are mean ± sd. *p* values are indicated. **c** Relative expression levels of *IL11* in CAF1 and CAF2 treated with oxaliplatin. *n* = 3 biologically independent experiments. Values are mean ± sd. *p* values are indicated. **d** Relative expression levels of *IL11* in CCD-18Co 6, 12, 90 and 180 days after oxaliplatin retrieval. *n* = 3 biologically independent experiments. Values are mean ± sd. *p* values are indicated. **e** Relative expression levels of *IL11* in CCD-18Co treated with oxaliplatin w/o TGF-beta pathway inhibitor (TBRi). *n* = 3 biologically independent experiments. Values are mean ± sd. *p* values are indicated. **f** Kaplan–Meier curve displays DFS for GSE39582 patients presenting low (blue; *n* = 136) or high (red; *n* = 383) expression levels of *IL11*. HR, CI and *p* value are indicated. **g** *IL11* levels in *n* = 1029 CRC patients classified by CMS subtypes (CMS1 *n* = 175; CMS2 *n* = 445; CMS3 *n* = 147, CMS4 *n* = 262). Central mark indicates the median, box extends from the 25th to 75th percentiles, whiskers represent the maximum and minimum data point. *p* values are indicated. **h** Kaplan–Meier curve displays DFS for GSE39582_CMS4 patients presenting low (blue; *n* = 26) or high (red; *n* = 95) expression levels of *IL11*. HR, CI and *p* value are indicated. **i** GSEA of CRC-IL11RS comparing relapsing (*n* = 13) to non-relapsing (*n* = 38) patients after CT in GSE14333. **j** Kaplan–Meier plot displays tumor initiation overtime in nude mice injected subcutaneously with 30,000 HT29-M6 control (blue; *n* = 10) or IL11-secreting cells (red; *n* = 10). *p* value is indicated. **k** Kaplan–Meier plot displays tumor initiation overtime in nude mice injected subcutaneously with 30,000 HT29-M6 control cells (dashed red; *n* = 12), TGF-beta secreting cells (red; *n* = 12), shIL11RA cells (dashed blue; *n* = 8) or with shIL11RA/TGF-beta-secreting cells (blue; *n* = 8). *p* value is indicated. CRC: HT29-M6; TFS tumor-free survival, DFS disease-free survival, HR hazard ratio, CI confidence interval, Ox oxaliplatin, Ct control, CT chemotherapy. Two-sided, unpaired *t*-test *p* values (*p*) are indicated for **b**–**e**, **g**. Log-rank (Mantel–Cox test) *p* values (*p*) are indicated for **f**, **h**, **j**, **k**. GSEA nominal *p* value (*p*) and FDR-adjusted *p* value are indicated for **i**. Source data are provided as a Source Data file.

The use of molecular markers guiding patient's treatment is crucial for bringing precision medicine into practice. However, no biomarkers of response to CT have been identified up until now. On the other hand, the clinical utility of molecular classifications based on whole-tumor transcriptomics remains limited due to gene expression variations arising from changes in cancer cells and stromal abundance[69,70]. Here, we report that high stromal expression of POSTN detected by IHC identifies CMS4 patients, associates with increased TGF-beta activity and predicts worse outcome independently of the tumor stromal load. In addition, low POSTN levels segregate a subset of CMS4 patients that will not face disease recurrence. We establish that POSTN protein is highly expressed before and/or after treatment in tumors unresponsive to therapy. Of note, not all treated tumors upregulate TGF-beta activity/POSTN expression upon CT. Indeed, tumors responsive to CT show low and stable levels of POSTN before and after treatment. The recent discovery of different subtypes of CAFs populating the tumors could explain these differences and further investigation may uncover specific CAF subtypes associated with response or resistance to CT. Nevertheless, our data argue that the evaluation of POSTN protein levels in patient samples before and after CT holds the potential to provide key information about primary or acquired resistance to therapy. Our findings further imply that the elevated expression of POSTN observed in CMS4 CRC could explain the primary resistance to CT associated with this molecular subtype. Alternatively, the increased expression of POSTN detected after CT in resistant tumors may indicate a transition upon treatment of the TME into a CMS4-like stroma in patients becoming unresponsive to therapy.

## Methods

### Study approval

Biological samples and clinical data were obtained under patient informed consent and approval of Clinical Research Ethics Committees (CEIC; 2016/6958/I, 2020/9113/I, 2020/9038/I) Parc de Salut MAR Biobank, IMIM, Spain. Informed consent authorizes the use of clinical information and biological surplus from diagnostic or therapeutic procedures for biomedical research projects. There was no active recruitment of patients for this study. There was no participant compensation. Experiments with mouse models were approved by the Animal Research Ethical Committee of Barcelona Biomedical Research Park and the Catalan government (CEEA-PRBB; FUE-2018-00801894).

### Generation of gene expression signatures and association with clinical parameters

To assess associations between gene expression profiles and clinical information, we used publicly available datasets: GSE17536[20], GSE39582[21], GSE14333[23] and GSE72970[22]. GSE17536 and GSE39582 contain respectively a pool of 177 and 519 CRC patients with clinical annotations. Patients from GSE39582 and GSE14333 were classified using the Random Forest method from CMSclassifier v.1.0.0 package in R v.3.5.1[8]. Patients with a CMS4 probability above 0.5 were selected to discard ambiguous results. A total of 121 (GSE39582_CMS4) and 10 samples (GSE14333_CMS4) were classified as CMS4. For GSE72970 and GSE14333, we studied respectively the subset of 32 advanced CRC patients and the subset of 51 stage III patients that had received standard adjuvant 5-fluorouracil plus oxaliplatin regimen. Data were downloaded from GEO microarray data repositories. Preprocessed series matrixes provided by the authors were used in the analyses.

To obtain the gene set activated by oxaliplatin in fibroblasts (aFib-RS), CCD-18Co cells were seeded at 80% confluence and treated with oxaliplatin (100 µM) for 6 days. For TGF-beta response signature in fibroblasts (Fib-TBRS), CCD-18Co cells were seeded at 60% confluence and treated with recombinant human TGF-beta 1 (Peprotech; 5 ng/ml) for 8 h. Gene expression profiles were measured in duplicate using PrimeView or HG-133+PM microarrays (Affymetrix). Analyses were performed with Transcriptome Analysis Console (TAC v.4.0) Software (Applied Biosystems) applying RMA summarization. aFib-RS includes genes upregulated two folds with *p* value <0.05 in oxaliplatin-treated cells. Fib-TBRS includes genes upregulated 1.5 folds with *p* value <0.05 in TGF-beta 1-treated cells. Gene signatures for hallmarks of P53 pathway, DNA repair and for cellular apoptosis were obtained from the Broad Institute database[24]. Senescence and senescence-associated secretory signatures (SenRS and SASP-S respectively) have been previously described[27,29]. For FAP (+) and (−) aFib-RS, Fib-TBRS and SASP-S subsets, FAP (+) specific probe set was obtained (FC > 1; *p* < 0.05; FDR < 0.01) using transcriptomic gene profiles of EpCAM (+), CD31 (+), CD45 (+) and FAP (+) FACS purified cell subsets retrieved from previous studies (GSE39396)[14]. Gene signatures are summarized in probe sets (g:Profiler v.2020-10-12) in Supplementary Data 1.

Gene Set Enrichment Analysis (GSEA v.4.1.0) was performed as previously described to obtain an enrichment score (ES), a normalized enrichment score (NES) which accounts for the size of the gene set being tested, a *p* value and an estimated False Discovery Rate (FDR)[71]. For single-sample Gene Set Enrichment Analysis (ssGSEA), ssGSEA module v.10.0.12 (GenePattern) was used as previously described[72]. ssGSEA projection provides data transformation from genes to gene-set thereby allowing to characterize biological processes and pathways rather than individual genes. Datasets were interrogated for generated signatures and correlation analyses between signatures were performed using Prism software (GraphPad v.8.0.1). We assessed signatures and single gene predictive significance on recurrence. Kaplan–Meier survival curves for patients with low and high average signature scores were obtained using Prism software (GraphPad v.8.0.1) and significance was assessed by log-rank (Mantel–Cox) test.

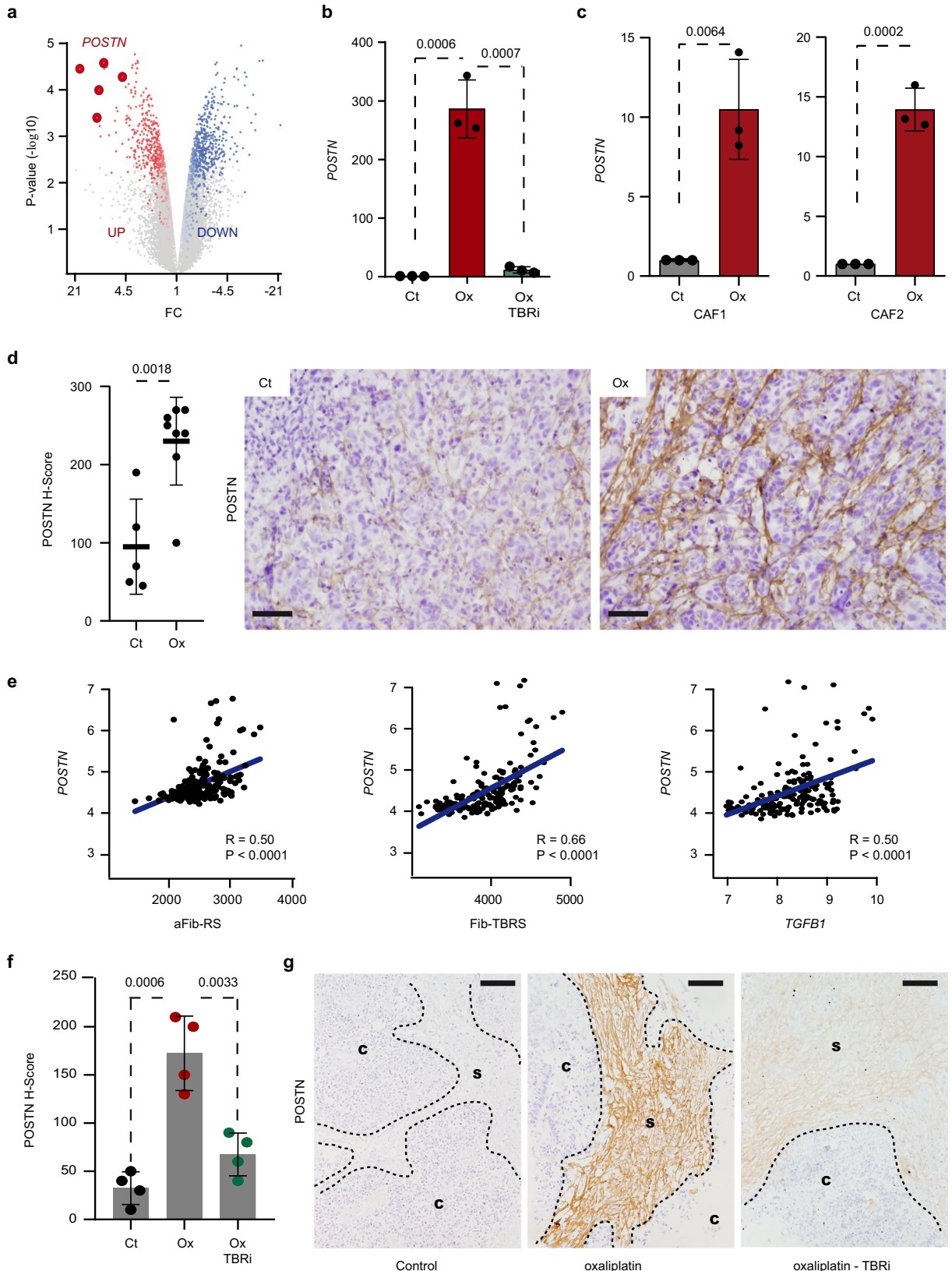

Optimal cutoff points were defined as the value giving the most significant split calculated by log-rank test using surv_cutpoint, survfit and coxph functions as implemented in Survival v.3.5-0 and Survminer v.0.4.9 R packages. Multivariate Cox regression model included available clinical variables as adjustment covariates. For CMS characterization, publically available merged expression data were obtained from Synapse repository (doi:10.7303/syn2623706) and analyzed as described elsewhere[8].

## Cell culture

MTOs were previously derived from primary tumors arising in genetically engineered mouse models with compound genetic alterations

**Fig. 5 | POSTN is marker of platinum-induced TGF-beta activity in CAFs.**
**a** Volcano plot displays genes upregulated and downregulated in fibroblasts treated with oxaliplatin (*n* = 2) compared to untreated ones (*n* = 2). *X* axis is the linear fold change and *Y* axis the −log10 *p* value of the ANOVA *p* values between conditions. *POSTN* probes are indicated. **b** Relative expression levels of *POSTN* in CCD-18Co treated with oxaliplatin w/o TGF-beta pathway inhibitor (TBRi). *n* = 3 biologically independent experiments. Values are mean ± sd. *p* values are indicated. **c** Relative expression levels of *POSTN* in CAF1 and CAF2 treated with oxaliplatin. *n* = 3 biologically independent experiments. Values are mean ± sd. *p* values are indicated. **d** Left panel: POSTN protein expression levels in tumors from MTO-injected mice treated with oxaliplatin (*n* = 8) compared to untreated control (*n* = 5).

Values are mean ± sd. *p* value is indicated. Middle and right panels: representative pictures of POSTN stained tumor sections. Scale bar: 50 μm. **e** Correlation between *POSTN* and aFib-RS (left panel), Fib-TBRS (center panel) or *TGFB1* (right panel) in *n* = 177 CRC tumors from GSE17536. Correlation values (*R*) and Spearman *p* values are indicated. **f** POSTN protein expression intensity in tumor samples obtained from untreated patients and cultured ex vivo with oxaliplatin in presence or absence of TGF-beta pathway inhibitor (TBRi). *n* = 4 biologically independent experiments. Values are mean ± sd. *p* values are indicated. **g** Representative POSTN immunostaining in explants from **f**. Scale bars: 100 μm. IU intensity unit, C cancer, S stroma, Ct control, Ox oxaliplatin. Two-sided, unpaired *t*-test *p* values (*p*) are indicated for **b**–**d**, **f**. Source data are provided as a Source Data file.

(*Apc, Kras, TrpS3, Tgfbr2*)[17]. For bioluminescence tracking, MTOs were infected with a lentivirus encoding an eGFP–firefly luciferase fusion reporter construct under the control of the *Pgk1* promoter[73]. For culture expansion, MTOs were embedded in basement-membrane extract (BME) medium (Cultrex BME Type 2, Amsbio) and cultured at 37 °C with 85–90% humidity, atmospheric $O_2$ and 5% $CO_2$ in advanced DMEM/F12 supplemented with 10 mM HEPES, Glutamax, B-27 without retinoid acid (all Life Technologies), 100 ng/ml recombinant NOGGIN and 50 ng/ml recombinant EGF (Peprotech). HUVEC (CRL-1730) and CCD-18Co (CRL-1459) were provided by the American Type Culture Collection (ATCC, USA). HT29-M6 cells were provided by the Cancer Cell Line Repository (CCLR) from MARBiobanc (Spain). Cell lines used in this study were not listed as known misidentified cell lines by the International Cell Line Authentication Committee. Cells were maintained in DMEM supplemented with 10% FBS and 1% glutamine. CRC patient-derived tumor organoids (PDOs)[33] were grown with PDO-specific media (Advanced DMEM/F12; 1x B-27 without retinoid acid; 1x Glutamax; 1x N-2, 1 mM N-Acetyl-L-cysteine and 50 ng/ml EGF) in BME (Reduced Growth Factor Basement Membrane Matrix, Type 2; Bio-Techne R&D Systems S.L.). PBMCs were purified from blood samples with Lymphoprep™ following manufacturers' protocols. To derive primary fibroblasts from patients, minced tissue samples were incubated with DMEM plus DNAse (10 μg/ml; Sigma-Aldrich) and Collagenase IV (100 U/ml; R&D) for 30 min. Pieces were seeded in plates and incubated at 37 °C in DMEM supplemented with 10% FBS, 1% glutamine, 20 ng/ml of fibroblast growth factor (FGF)−2 (Peprotech), P/S and normocin. Once fibroblasts adhered to the plate, pieces were removed. Fibroblasts were grown using standard methods at 37 °C and 5% $CO_2$.

To compare the biological activity of oxaliplatin after a 3-day treatment, half maximal effective concentration ($EC_{50}$) of oxaliplatin was defined for each cell subtype. In order to compare tumor cells response to CT over time, cultured cells were treated with oxaliplatin (100 μM) for 12 days. For pre-stimulation experiments, each cell subtype was treated for 6 days with its own oxaliplatin concentration $EC_{50}$ in order to avoid complete cell death. Media were changed every 3 days. For conditional media (CM) experiments, cultured cells were treated as mentioned above. After 6 days, oxaliplatin-containing media were discarded and cell cultures were washed 3 times with HBSS (Lonza) to ensure the removal of residual free CT. Next, fresh DMEM supplemented with 0.2% FBS was added to the culture. CM was collected after 48 h. To compare oxaliplatin and CPP-OX effect, fibroblasts were treated with 3 μM of each drug. When indicated, cells were treated with 5 μM TGFBR1 receptor inhibitor (SB431542; Life Science, Sigma-Aldrich).

pReceiver-Lv105 Expression plasmids containing the cDNA sequences of interest under the control of a CMV promoter were used to generate IL11 and POSTNi4 overexpressing cells (Genecopeia) by lentiviral infection. CRC cells secreting active TGF-beta 1 were generated after infection with a lentiviral vector (FUW) encoding the biologically active form of human TGF-beta 1. Empty vector was used to generate control cells. Knockdown experiments were conducted by lentiviral infection using short hairpin (sh) RNA targeting IL11RA (shIL11RA) (Sigma-Aldrich). Non-targeting shRNA sequence was used

as control (shCt) (Sigma-Aldrich). Genetic engineering efficiency was confirmed by qRT-PCR.

For bioluminescent tracking, cancer cells were infected with a fusion protein reporter construct encoding firefly luciferase. For cell survival experiments, XTT assay kit (Biological Industries) was used following manufacturer's protocol. Alternatively, cell survival was measured by bioluminescent tracking or by crystal violet assay as described elsewhere[74]. To detect senescent cells, ß-galactosidase staining kit was used as described in manufacturer protocol (Cell Signaling Technology®).

## Clinical material

For LA-ICP-MS analyses (Fig. 1e, f and Supplementary Fig. 2), FFPE biological samples were obtained from eight CRC patients previously treated with oxaliplatin-based CT (5 males; 3 females; median age at diagnosis: 56 years). For POSTN and stroma analyses in CMS (Fig. 6a–f and Supplementary 7f–i), we used FFPE tumor samples from 67 males and 42 females (median age at diagnosis: 71 years). For ex vivo experimentation (Fig. 5f, g), freshly resected primary tumor samples from two male and two female untreated CRC patients over 50 years old were collected and cultured at 37 °C/5% $CO_2$ in RPMI supplemented with 10% FBS, P/S and normocin[75]. Samples were treated ex vivo with oxaliplatin (100 μM) for 48 h and compared to untreated control from the same patient. When indicated, explants were treated with 5 μM TGFBR1 inhibitor (SB431542; Life Science, Sigma-Aldrich). For response to CT analyses (Figs. 3e and 6g, h), FFPE tumor samples were obtained from 28 CRC patients before and after oxaliplatin-based CT (16 males; 12 females; median age at diagnosis: 74 years). All samples were collected within the usual clinical practice and were utilized in this study per availability. Clinical information was anonymized by medical doctors collaborating to the project. International standards of Ethical Principles for Medical Research Involving Human subjects (code of ethics, Declaration of Helsinki, Fortaleza, Brazil, October 2013) were followed in accordance with legal regulations on data confidentiality (Organic Law 3/2018 -December the 5th- on the Protection of Personal Data and Digital Rights Guarantee) and on biomedical research (Law 14/2007 -July the 3rd-).

## Laser ablation inductively coupled plasma mass spectrometry (LA-ICP-MS)

LA-ICP-MS analyses were performed in 6 μm-thick sections of paraffin-embedded tumor samples previously stained with Hematoxylin & Eosin (H&E). Areas of interest were identified by expert pathologist. Laser ablation was achieved with an Analyte G2 instrument (Photonmachines Inc.). Of note, LA-ICP-MS results in the destruction of the tissue thus preventing further analysis of the same area. Levels of isotopes [194]Pt and [195]Pt were recorded with an ICP-QMS 7700 instrument (Agilent). Data were correlated with H&E to identify histological features at the microscopic level. Origin v.9.5 (OriginLab Corp.) and ImageJ v.1.53i were used to perform analyses and to generate Pt distribution maps.

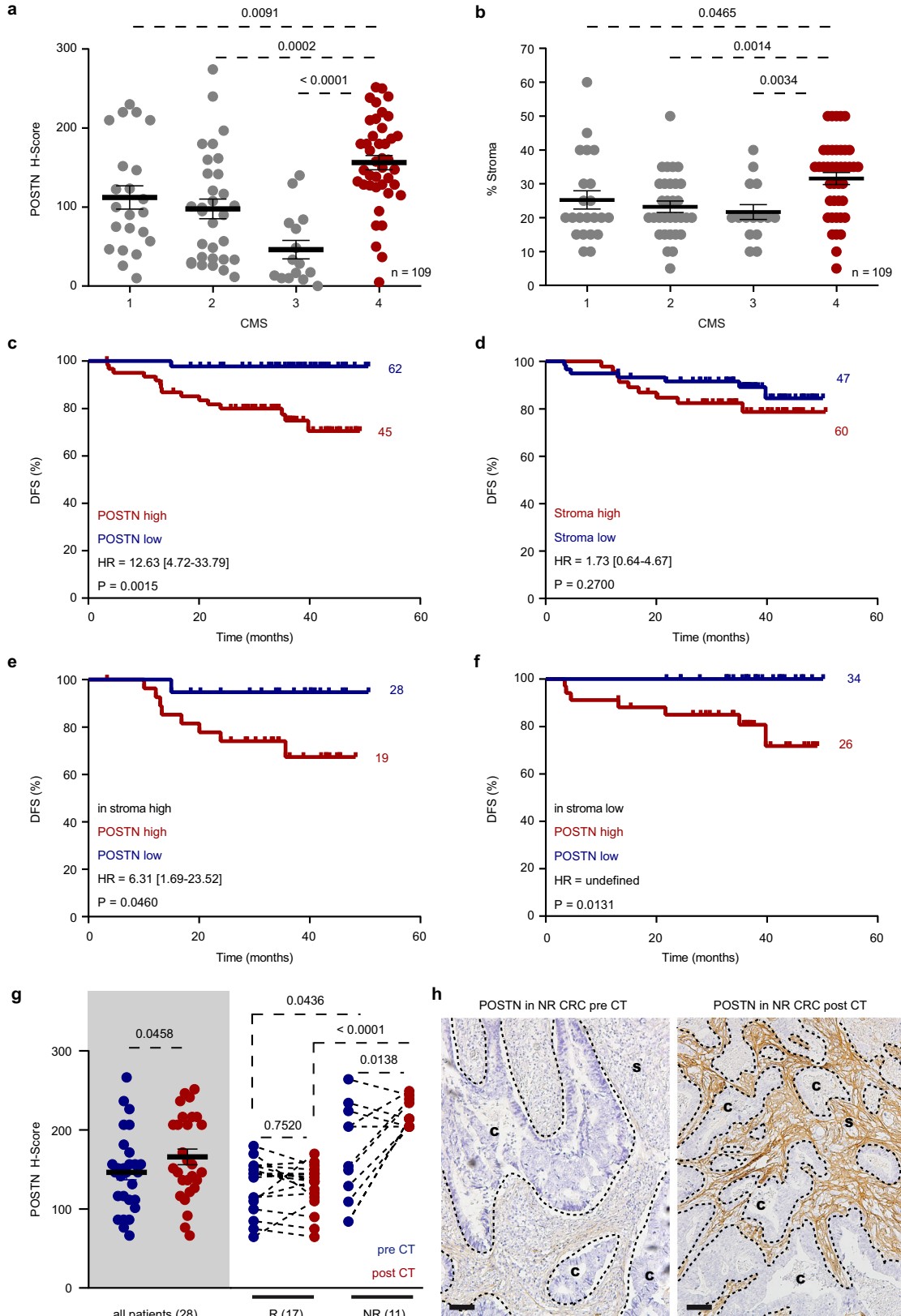

**Inductively coupled plasma mass spectrometry (ICP-MS)**

Cells were exposed to 3 µM of oxaliplatin for 4 h. After incubation, cells were washed with PBS and trypsinized. The cell suspension was centrifuged at $200 \times g$ for 5 min at 4 °C. Cell pellets were suspended in 300 µl of mQ $H_2O$ and sonicated during 30 min. Cell suspension was transferred into a Teflon reactor with 300 µl of HNO3 65% and mineralized at 90 °C for 18 h. The samples were diluted with mQ $H_2O$ until 2% $HNO_3$ concentration was reached. Platinum determination was performed with an Agilent 7500ce Series Inductively Coupled Plasma-Mass Spectrometer and normalized by protein quantity.

**Fig. 6 | POSTN is a stromal marker of resistance to chemotherapy. a** POSTN protein expression intensity in IHC-CRC cohort (CMS1 *n* = 22; CMS2 *n* = 31; CMS3 *n* = 15, CMS4 *n* = 41). Values are mean ± sem. *p* values are indicated. **b** Percentage of stroma in IHC-CRC cohort (CMS1 *n* = 22; CMS2 *n* = 31; CMS3 *n* = 15, CMS4 *n* = 41). Values are mean ± sem. *p* values are indicated. **c** Kaplan–Meier curve displays DFS of CRC patients in IHC-CRC cohort presenting low (blue; *n* = 62) or high (red; *n* = 45) protein expression of POSTN. HR, [CI], *p* value are indicated. **d** Kaplan–Meier curve displays DFS of CRC patients in IHC-CRC cohort presenting low (blue; *n* = 47) or high (red; *n* = 60) stromal content. HR, [CI], *p* value are indicated. **e** Kaplan–Meier curve displays DFS of CRC patients in IHC-CRC cohort presenting low (blue; *n* = 28) or high (red; *n* = 19) protein expression of POSTN in tumors with high stromal content. HR, [CI], *p* value are indicated. **f** Kaplan–Meier curve displays DFS of CRC patients in IHC-CRC cohort presenting low (blue; *n* = 34) or high (red; *n* = 26)

protein expression of POSTN in tumors with low stromal content. HR, [CI], *p* value are indicated. **g** Left gray panel: protein expression levels of POSTN in all (*n* = 28) paired tumor samples collected before (blue) and after (red) CT. Right panel: protein expression levels of POSTN in tumor samples categorized by patients responsive (R; *n* = 17) and unresponsive (NR; *n* = 11) to therapy. Values are mean ± sem. *p* values are indicated. **h** POSTN detection by IHC in paired tumor samples from a NR patient collected before (left panel) and after CT (right panel), representative of *n* = 11 patients. Scale bars: 100 μm. IU intensity unit, C cancer, S stroma, CMS consensus molecular subtype, CT chemotherapy, HR hazard ratio, CI confidence interval, NR unresponsive to CT, R responsive to CT, DFS disease-free survival. Two-sided, unpaired (**a**, **b**) and paired (**g**) *t*-test *p* values (*p*) are indicated. Log-rank (Mantel–Cox test) *p* values (*p*) are indicated for **c**–**f**. Source data are provided as a Source Data file.

## Western blot

Protein extracts were obtained by lysing cells in RIPA buffer (Tris HCl 50 mM ph 7.4, NP-40 1%, Na-Deoxycholate 0.25%, NaCl 150 mM, EDTA 1 mM). Protein concentration was measured using the Bio-rad kit Protein Assay. Proteins were separated by SDS gel electrophoresis and transferred to PVDF membrane (Millipore). Antibodies (Supplementary Table 3) were incubated o/n at 4 °C in 2% BSA TBS-Tween 0.1% (1/250 dilution). Anti-β-Actin antibody (ab20272, Abcam) was incubated 1 h at 4 °C in 2% BSA TBS-Tween 0.1% (1/30.000 dilution). Secondary antibodies (1/2.000 dilution) coupled to peroxidase were incubated for 1 h at RT. Membranes were washed in TBS-Tween 0.1%. Immuno-complexes were detected using Immobilon Western HRP (Millipore). Uncropped and unprocessed scans are shown in Source Data file (for Fig. 7e) and in Supplementary Fig. 9 (for Supplementary Figs. 4h, 5b and 6d).

## Immunohistochemistry

Immunostainings were carried out on 3 μm-thick tissue sections according to standard procedures. Briefly, after antigen retrieval, samples were blocked with Peroxidase-Blocking Solution (Dako, S202386) for 10 min at RT. Primary antibodies (please refer to Supplementary Table 3) were incubated o/n at 4 °C. Slides were washed with EnVision™ FLEX Wash Buffer (Dako, K800721). Secondary antibody was incubated for 45 min at RT. For anti-rat Biotin Donkey IgG, additional 20 min incubation at RT with streptavidin HRP (Sigma, S2438, 1/1000 dilution) was performed. Samples were developed using 3,3'-diaminobenzidine, counterstained with hematoxylin and mounted. Staining analyses were performed with either QuPath software v.0.3.2[76] or H-Score[77]. In more details, H-Score representing overall stromal POSTN protein levels was assessed for each sample by expert pathologist using intensity scores (from 0 to 3) and intensity scores frequencies (from 0 to 100%). H-Score was calculated as follow: (% of scored 1 stromal area) × 1 + (% of scored 2 stromal area) × 2 + (% of scored 3 stromal area) × 3.

## Quantitative RT-PCR

Reverse transcription was performed using High Capacity cDNA Reverse Transcription Kit (Applied Biosystems). Quantitative PCR was achieved using TaqMan and SYBR green reagents (Applied Biosystems) following manufacturer's instructions in a 7900HT Fast Real-Time System (Applied Biosystems). TaqMan assays (Applied Biosystems; *CDKN1A*: Hs00355782_m1, *CDKN2A*: Hs00923894_m1, *TGFB1*: Hs00998133_m1, *IL11*: Hs01055414_m1, *POSTN*: Hs00170815_m1, *IL1B*: Hs01555410_m1, *CXCL8*: Hs00174103_m1, *GDF15*: Hs00171132_m1, *B2M*: Hs99999907_m1, *GAPDH*: Hs99999905_m1, *PPIA*: Hs99999904_m1) and SYBR Green primers (*POSTNi1* (F: GTGATTGAAGGCAGTCTT CAGCC; R: CTCCCTGAAGCAGTCTTTTA), *POSTNi2* (F: AATCCCCGT GACTGTCTATAAGCCA; R: CTCCCTGAAGCAGTCTTTTA), *POSTNi3* (F: AATCCCCGTGACTGTCTATAGACC; R: TCCTCACGGGTGTGTCTTCT), *POSTNi4* (F: AATCCCCGTGACTGTCTATAAGCCA; R: TCCTCACGGGT

GTGTCTTCT), *POSTNi5* (F: AATCCCCGTGACTGTCTATAGACC; R: CTCCCTGAAGCAGTCTTTTA), *POSTNi6* (F: AATCCCCGTGACTGTCTA TAGTCCT; R: CTCCCTGAAGCAGTCTTTTA), *POSTNi7* (F: AATCCCCGT GACTGTCTATAGTCCT; R: TCCTCACGGGTGTGTCTTCT), *POSTNi8* (F: GTGATTGAAGGCAGTCTTCAGCC; R: TCCTCACGGGTGTGTCTTCT), *GAPDH* (F: GGAGTCAACGGATTTGGTCGTA; R: GGCAACAATATCCA CTTTACCAGAGT) were used. Quantitative RT-PCR Data were analyzed using SDS v2.4 software (Applied Biosystems). For pre/post CT tumor patient analysis, samples with similar cancer cells/ stromal cells ratio were selected. RNA was extracted from FFPE section following manufacturer's instruction (RNeasy FFPE Kit, QIAGEN).

## In vivo studies

Animals were maintained in specific pathogen-free conditions with controlled temperature/humidity (22 °C/55%) environment on a 12-h light-dark cycle and with standard diet and water ad libitum. Sample size was predetermined empirically according to previous experience using the same strains and treatments. The general condition of animals was monitored using animal fitness and weight controls by authors, facility technicians and by an external veterinary scientist responsible for animal welfare.

For intracaecum CRC model, $0.1 \times 10^6$ cells in 70% BME were injected with a 30 G syringe under binocular guidance into the submucosal wall of the distal caecum of 7–9 weeks old female *C57BL/6J* (strain #:C57BL/6JRj) mice purchased from Janvier Labs. Mice bearing tumors were randomly assigned to treatment (*n* = 8) and control groups (*n* = 5). Treated mice were injected IP with oxaliplatin (12 mg/kg) 96 h and 24 h before tumor resection. Tumor growth was monitored using bioluminescence. For subcutaneous CRC model, cells were injected subcutaneously in 5–6 weeks old female NSG *NOD.Cg-Prkdcscid Il2rgtm1Wjl/SzJ* (Strain #:005557) or nude *NU/J* (Strain #: 002019) mice (Jackson Laboratories). Subcutaneous tumor appearance was assessed by palpation. For resistance to CT experiments, macroscopic tumor-bearing mice (average tumor size 50 mm³) were injected IP once per week with oxaliplatin (12 mg/kg). Tumor volume was measured twice a week by caliper until sacrifice. Maximal tumor burden permitted is 1500 mm³. In some cases, this limit has been exceeded the last day of measurement and mice were immediately euthanized. In one case, the limit was exceeded with the approval of veterinarians. At experimental end point, mice were euthanized in a chamber with saturated $CO_2$ atmosphere. Euthanasia was confirmed by cervical dislocation.

## Statistics

Sample size was chosen following previous experience in the assessment of experimental variability (generally all measurements were performed with *n* ≥ 3 biological replicates). Statistical analyses of between-group differences were performed using Student's *t* test (Graphpad Prism 8.0.1). Two-tailed *p* values <0.05 were considered significant.

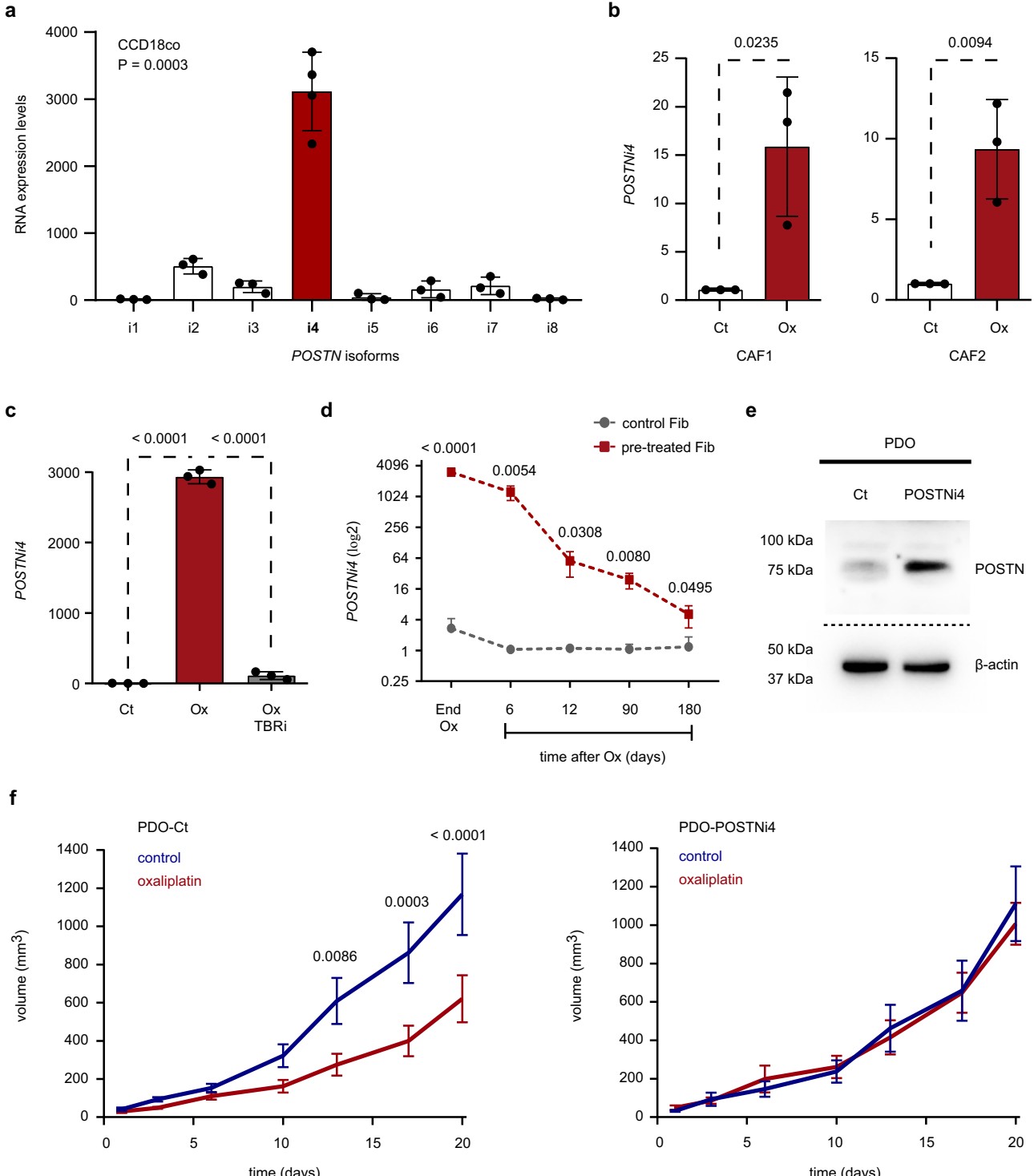

**Fig. 7 | Platinum-induced expression of POSTN isoform 4 in the tumor stroma enhances resistance to treatment. a** Relative expression levels of *POSTN* isoforms in CCD-18Co treated with oxaliplatin. $n = 4$ and $n = 3$ biologically independent experiments for *POSTNi4* and for *POSTNi1-3, 5-8* respectively. Values are mean ± sd. *p* value is indicated. **b** Relative expression levels of *POSTNi4* in CAFs treated with oxaliplatin. $n = 3$ biologically independent experiments. Values are mean ± sd. *p* values are indicated. **c** Relative expression levels of *POSTNi4* in CCD-18Co treated with oxaliplatin w/o TGF-beta pathway inhibitor (TBRi). $n = 3$ biologically independent experiments. Values are mean ± sd. *p* values are indicated. **d** Relative expression levels of *POSTNi4* in CCD-18Co 6, 12, 90, 180 days after oxaliplatin retrieval. $n = 3$ biologically independent experiments. Values are mean ± sd. *p* value is indicated. **e** POSTN protein levels in genetically engineered PDOs with upregulated expression of POSTNi4 compared to control cells. Bottom panel shows β-Actin protein levels as normalization control. Representative of $n = 3$ biologically independent experiments. **f** Growth kinetics upon oxaliplatin treatment of tumors derived from subcutaneous injection into NSG mice of PDOs-Ct or POSTNi4-secreting PDOs. Left panel: PDOs-Ct injected mice tumors treated (red; $n = 10$) or non-treated tumors (blue; $n = 9$). Right panel: POSTNi4-PDOs injected mice tumors treated with oxaliplatin (red; $n = 9$) or non-treated tumors (blue; $n = 7$). Values are mean ± sem. *p* values are indicated. PDO patient-derived tumor organoids, Fib CCD-18Co, Ct control, Ox oxaliplatin. Two-sided, unpaired *t*-test *p* values (*p*) are indicated for **a**–**d**, **f**. Source data are provided as a Source Data file.

**Reporting summary**

Further information on research design is available in the Nature Portfolio Reporting Summary linked to this article.

## Data availability

The transcriptomic datasets generated for this study have been deposited in NCBI GEO repository under the accession numbers GSE181020 and GSE181026. Hallmarks gene signatures were obtained from the Broad Institute database (https://www.gsea-msigdb.org/). Publicly available merged expression data were obtained from Synapse repository (doi:10.7303/syn2623706). GSE39396, GSE17536, GSE39582, GSE72970 and GSE14333 datasets used in this study are publicly available in the NCBI GEO database. The remaining data are available within the Article, Supplementary Information or Source Data file. Source data are provided with this paper.

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

## Acknowledgements

This work has been supported by grants from Fundación científica AECC -Asociación Española contra el Cáncer- (GCAEC20030CERV) to A.Ce., from Instituto de Salud Carlos III (ISCIII) co-funded by the European Union (CP16/00151, PI17/00211, PI20/00011; Spanish Ministry of Economy and Competitiveness) to A.Ca. and PI20/00625 to P.N., from la Caixa Foundation (LCF/PR/HR19/52160018) and MICINN (PID2020-119917RB-I00) to E.B., from Spanish Ministerio de Economia y Competitividad (MINECO) and FEDER funds (PID2019-104948RB-I00) to R.R.G. This work was supported by Grant PT20/00023, funded by Instituto de Salud Carlos III (ISCIII) and co-funded by the European Union, and the Xarxa de Bancs de tumors sponsored by Pla Director d'Oncologia de Catalunya (XBTC). A.Ca. is the recipient of funding from the Instituto de Salud Carlos III co-funded by the European Union (MS16/00151; CPII21/

00012). J.L. is the recipient of a Junior Clinician fellowship from Fundación científica AECC (CLJUN19004LINA).

## Author contributions

A.Ca. conceived and designed the study. J.L., A.S.A., A.R.B. and E.I.R. set up the experimental models and performed treatment analyses. N.T., M.O. and N.S. managed patients' data and samples collection. C.M., A.Ce., J.T. and E.E. performed clinical analyses. C.M.C. and M.I. performed histopathological analyses. J.L., A.S.A., A.R.B. and M.Z. performed the biomolecular analyses. N.M.R. and P.N. performed proteomics analyses. D.L.R. and M.V. performed chemical analyses. A.M. and J.P. performed LA-ICP-MS analyses. M.Ga. and A.S.A. performed immunostaining experiments. R.R.G., E.B., X.H.M., J.B.R., A.C.S. and M.Gu. designed and performed in vivo experiments. J.L., A.Ce., C.M., E.B., R.R.G., J.P. and A.Ca. interpreted and discussed the results. A.Ca. and J.B.R. performed transcriptomic analyses. A.Ca. wrote the manuscript. All authors approved this manuscript for publication.

## Competing interests

A.Ce. declares institutional research funding from Genentech, Merck Serono, Bristol Myers Squibb, Merck Sharp & Dohme, Roche, Beigene, Bayer, Servier, Lilly, Novartis, Takeda, Astellas, Takeda and Fibrogen; and advisory board or speaker fees from Amgen, Merck Serono, Roche, Bayer, Servier and Pierre Fabre in the last 5 years. J.T. reports personal financial interest in form of scientific consultancy role for Array Biopharma, AstraZeneca, Bayer, Boehringer Ingelheim, Chugai, Daiichi Sankyo, F. Hoffmann-La Roche Ltd, Genentech Inc, HalioDX SAS, Hutchison MediPharma International, Ikena Oncology, Inspirna Inc, IQVIA, Lilly, Menarini, Merck Serono, Merus, MSD, Mirati, Neophore, Novartis, Ona Therapeutics, Orion Biotechnology, Peptomyc, Pfizer, Pierre Fabre, Samsung Bioepis, Sanofi, Scandion Oncology, Scorpion Therapeutics, Seattle Genetics, Servier, Sotio Biotech, Taiho, Tessa Therapeutics and TheraMyc. Stocks: Oniria Therapeutics and also educational collaboration with Imedex/HMP, Medscape Education, MJH Life Sciences, PeerView Institute for Medical Education and Physicians Education Resource (PER). N.T. declares the following: Advisory Role: Merck Serono, Guardant Health, Speaking: Amgen, Servier, Pfizer, Merck Serono, ESMO, SEOM. C.M. reports personal financial interest in form of scientific consultancy role for Amgen, Biocartis, F. Hoffmann-La Roche Ltd, Genentech Inc, Merck Serono, Pfizer, Pierre Fabre, Sanofi, also educational collaboration with Amgen, Guardant Health, Merck Serono. The remaining authors declare no other competing interests.

## Additional information

[1]Cancer Research Program, Hospital del Mar Medical Research Institute (IMIM), Barcelona, Spain. [2]Department of Medical Oncology, Catalan Institute of Oncology (ICO), Barcelona, Spain. [3]Department of Physics, Faculty of Science, University of Oviedo, Oviedo, Spain. [4]Institute for Research in Biomedicine (IRB Barcelona), The Barcelona Institute of Science and Technology (BIST), Barcelona, Spain. [5]Centro de Investigación Biomédica en Red de Cáncer (CIBERONC), Madrid, Spain. [6]Department of Pathology, Hospital Clínico Universitario, INCLIVA Biomedical Research Institute, University of Valencia, Valencia, Spain. [7]Department of Medical Oncology, Hospital Clínico Universitario, INCLIVA Biomedical Research Institute, University of Valencia, Valencia, Spain. [8]Medical Oncology Department, Hospital del Mar, Barcelona, Spain. [9]Vall d'Hebron Hospital Campus and Institute of Oncology (VHIO), Barcelona, Spain. [10]Institute of Biomedical Research of Barcelona (IIBB-CSIC), Barcelona, Spain. [11]Institut d'Investigacions Biomèdiques August Pi Sunyer (IDIBAPS), Barcelona, Spain. [12]Cancer Research Program, Hospital del Mar Medical Research Institute (IMIM), Unidad Asociada IIBB-CSIC, Barcelona, Spain. [13]Institució Catalana de Recerca i Estudis Avançats (ICREA), Barcelona, Spain. ✉e-mail: acalon@imim.es

