## [Peer Review File · Nature Communications]

Long-term platinum-based drug accumulation in cancer-associated fibroblasts promotes colorectal cancer progression and resistance to therapyREVIEWER COMMENTS

Reviewer #1 (Remarks to the Author): with expertise in colorectal cancer, cancer associated fibroblasts

Linares et al describe a mechanism for therapeutic resistance to chemotherapy in colon cancer, mediated by CAFs through accumulation of platinum-based drugs. They began their study by assessing platinum distribution in FFPE tumor samples from CRC patients, and found that oxaliplatin accumulated predominantly in the stroma, up to months following treatment, which was significantly longer than the accumulation in cancer cells. To mechanistically dissect this, they measured oxaliplatin accumulation in cancer cells and colon fibroblasts, in vitro, and found that fibroblasts were significantly more resistant to this treatment compared to cancer cells. They then coinjected fibroblasts with cancer cells into nude mice, and found that platinum-treated fibroblasts induced faster growing tumors compared to non-treated fibroblasts.

They performed microarray-based transcriptional profiling to determine a gene signature associated with platinum-resistance in fibroblasts and used this signature to interrogate patient datasets. They found that expression of these genes is associated with poor survival and resistance to treatment. They performed additional bioinformatic analyses and chose to focus on TGF-beta and IL11. They found that IL11 expression was induced in colon fibroblasts as well as patient CAFs in response to platinum treatment and this could be inhibited by tgf-beta inhibitors. They overexpressed IL11 in cancer cells injected into nude mice and found that this increased tumor progression. Finally they show that POSTN expression in fibroblasts is also induced following platinum treatment and suggest that POSTN could be a potential biomarker for response to therapy.

Overall this is an interesting study, of a timely topic – the role of CAFs in mediating drug resistance. Nevertheless there are conceptual and technical limitations to this study that must be addressed to support the far-reaching conclusions made by the authors. In particular, this reviewer is not convinced that CAFs are the ones mediating resistance in patients (rather than other cells in the TME) or that this mechanism contributes to resistance in patients.

Specific comments:

1. Currently there is insufficient evidence for accumulation of platinum in CAFs rather than other stromal cells such as endothelial, immune etc. The authors show patient samples in Figure 1, but no costaining for CAF markers. Perhaps this is mediated by other cells in the TME? More images, more patients and most importantly costaining are needed to make this point convincing.
2. Following up on this, the authors treat fibroblasts in-vitro with platinum and show that the CM and/or cells induce resistance. Would this not have been achieved by treatment of any other cell type (again, endothelial and immune but also others - even cancer cells) with platinum?
3. Similarly, the authors must show that changes in TGF-beta and IL11 signaling indeed occur in CAFs and not in other cells in the TME (costaining in patients or mice). In fact they show the contrary – that modulation of IL11 in cancer cells created the same effect. It is possible that treating cancer cells with CM from epithelial or endothelial cells pretreated with platinum, would have yielded similar effects to the CAFs. Moreover, how do the authors exclude the effect of residual chemo in the medium?
4. Moreover, the sequencing data is from bulk, so once again these changes in gene expression may not be specific to CAFs. In fact the pathways most differentially expressed are indeed generic to drug treatment - P53 pathway, DNA repair and cellular apoptosis. The authors focus on TGF-beta, but changes in TGF-beta are not specific to drug treatment but rather a hallmark of the transition from normal fibroblasts to myofibroblastic CAFs. Certainly there are genes that are better hallmarks of SASP than TGF-beta.
5. The data regarding POSTN is nice and looks like it is staining CAFs but again no costaining with CAF markers is shown. Also, how was the quantification done?
6. The authors mention the use of combination treatment today rather than platinum alone which brings the question of relevance of these findings, especially since the mouse study was done in immune-deficient mice, thus already limiting the TME responses.

Minor comments:

1. The authors should describe in the intro the CRC subtypes (row 65)
2. Many of the graphs show bars and not dots and it is hard to assess how the replicates distribute.
3. Figure 2F – how was high vs low determined? It is not the median, so what was the criterion?

Reviewer #2 (Remarks to the Author): with expertise in colorectal cancer, periostin

In this manuscript, Linares et al found that platinum-based drugs accumulate in cancer-associated fibroblasts (CAFs) long time after treatment in colorectal cancer (CRC) patients. Oxaliplatin accumulation activates the TGF- β pathway in CAFs, which induces the expression of IL-11 and periostin (POSTN). POSTN isoform 4 is the main splicing variant upregulated in CAFs and is a marker of poor prognosis. The pro-oncogenic secretory phenotype in CRC fibroblasts induced by oxaliplatin accumulation promotes CRC progression and resistance to chemotherapy. The authors discover an interesting and important finding that platinum accumulation in CAFs promotes CRC progression, which may contribute to CRC treatment. However, there is lots of correlation analyses in this manuscript but the directly regulation mechanism is not very clear. Multiple other concerns are needed to be addressed:

Major points:

1. The authors used nude mice subcutaneously injected CRC cells and CAFs to do the in vivo studies. It is better to add a spontaneous CRC mouse model (APC mim/+) or AOM/DSS-induced CRC mouse model, which can receive the treatment with platinum-based drugs to detect the activation of TGF- β , upregulation of IL-11 and POSTN in CAFs and the related regulation mechanism.
2. Based on the authors' results, oxaliplatin combined with POSTN blockade supposes to achieve better therapeutic efficacy for those patients unresponsive to oxaliplatin chemotherapy. It will be very interesting to design animal studies to verify the authors' hypothesis.
3. Fig. 2d show the Kaplan-Meier plot of nude mice injected subcutaneously with CRC cells alone, co-injected with CCD18co non-treated or pre-treated with oxaliplatin. What about tumor size? Does CCD18co pre-treated with oxaliplatin influence tumor growth and tumor cell proliferation?
4. Based on Fig. 3h the Fib-TBRS was significantly increased in oxaliplatin-treated CCD-18co, the authors concluded that "an autocrine activation of the TGF-beta pathway in fibroblasts" (Line 179). Previous evidence show that TGF-beta is mainly produced by colorectal tumor cells in CRC. Therefore, representative protein levels (TGF- β and/or p-Smad3, p-Smad4) need to be detected in CAFs and in tumor tissues.
5. Fig. 4b-e shows the transcription levels of IL-11 upregulated after oxaliplatin treatment, and the protein levels of IL-11 should also be tested.
6. In Fig. 4e TGF- β pathway inhibitor partially reduces the expression levels of IL-11 induced by oxaliplatin treatment. What other signaling pathways other than the TGF- β pathway also regulate IL-11 expression after oxaliplatin treatment? In addition, how does oxaliplatin accumulation activate the TGF- β pathway in CAFs?
7. Fig. 5c-d shows the transcription levels of POSTN upregulated in CAFs after oxaliplatin treatment, and the protein levels of POSTN should also be tested.
8. In Fig. 5f TGF- β pathway inhibitor partially reduces the expression levels of POSTN induced by oxaliplatin treatment. Apart from the TGF- β pathway, is POSTN expression regulated by other pathways after oxaliplatin treatment?

Minor points:

1. Fig. 7f and Fig. 7g are the same experiments of different groups. Combine Fig. 7f and Fig. 7g into one graph.

Reviewer #3 (Remarks to the Author): with expertise in colorectal cancer, gene signatures

CMS4 subtype colorectal cancer is characterized by poor clinical outcome even after standard chemotherapy with 5-fluorouracil/leucovorin with or without oxaliplatin. In a retrospective analysis of NSABP clinical trial C-07 which evaluated the efficacy of oxaliplatin added to 5-FU/leucovorin, patients with CMS4 subtype colon cancer did not benefit from the addition of oxaliplatin. CMS4 tumors are characterized by high stromal content and resulting gene expression signature mimicking epithelial mesenchymal transition – however, investigators have demonstrated that EMT and TGF-beta signature of CMS4 tumors originate from stromal fibroblasts rather than cancer cells, although conflicting data do exist. Mechanism of oxaliplatin resistance of CMS4 colon cancer remained unknown.

Linares et al attempted to identify the mechanism of oxaliplatin resistance of CMS4 tumors. First, they checked the presence of oxaliplatin in the tumor tissue after therapy in 31 residual colorectal cancer. This experiment demonstrated a presence of residual oxaliplatin only the stromal cells mainly cancer associated fibroblasts (CAF). In vitro experiments were conducted using CAFs to demonstrated that TGF-beta signaling was augmented by the retained oxaliplatin resulting in secretion of multiple factors associated with poor prognosis, such as IL11, a prometastatic cytokine. Periostin (POSTN), especially a specific isoform POSTNis4, was identified as a marker that is induced by oxaliplatin treated CAFs and forced expression of POSTNis4 in colon cancer organoids resulted in oxaliplatin resistance. POSTN overexpression in the tumor stroma was associated with poor clinical outcome and was associated with CMS4 subtype.

The study successfully identified the mechanism of oxaliplatin resistance in CMS4 colon cancer and suggest potential ways of overcoming chemo-resistance through intervention of molecules such as IL11 and POSTN secreted by CAFs.

Currently there is no widely used clinical test to identify CMS4 colon cancer subtype. This study identifies POSTN immunohistochemistry as a potential clinical surrogate marker of CMS4 subtype.

Soonmyung Paik, MD
Professor
Yonsei University College of Medicine

Reviewer #4 (Remarks to the Author): with expertise in colorectal cancer, gene signatures, CRC stroma

In this study by Linares and colleagues, the authors build on new elemental imaging observations of oxaliplatin accumulation in the stromal components of tumour tissue, to develop and characterise oxaliplatin activated fibroblasts. Molecular signatures induced in these activated fibroblasts are aligned to clinical correlates, where they are associated with poor outcomes and limited response to therapy. The authors eloquently demonstrate the essential nature of TGFb and POSTN signalling in this molecular axis.

The authors have presented an interesting and important piece of work, and have utilised existing models and molecular data to demonstrate the associations with the transcriptional targets upregulated in oxaliplatin activated stroma. These new data fit with the group's prior publications into the role and importance of the stroma and TGFb signalling in colorectal cancer, and will make a valuable contribution to the field. The finding of oxaliplatin accumulation is certainly of mechanistic importance, and exploitation of this will inevitably have therapeutic relevance. There is limited mechanistic data presented as to how this accumulation happens, or how it can either be prevented or reduced.

The main data in the paper looks to utilise the signature generated from oxaliplatin treated fibroblasts, which display increased TGFb signalling, therefore my main question is to what extent that this new signature provides more useful information above the numerous stromal signatures that exist, and the TGFb-stimulated fibroblast signatures previously developed/tested by the authors. I have suggested a

number of points below that I hope will give the reader additional confidence in the data presented, and have also asked for some specific analyses that it is hoped will enable the authors to demonstrate if this molecular insight oxaliplatin-activated fibroblast signalling provides additional biomarker or mechanistic value beyond the groups previously defined fibroblast signatures.

Main essential points:

Point 1 - The authors state that, in line 146-148: “Gene set enrichment analysis (GSEA) showed that aFib-RS was significantly upregulated in tumors from unresponsive (Figure 2g) and relapsing patients (Figure 2h), thus indicating an association between the gene program activated in fibroblasts upon oxaliplatin treatment and the lack of benefit from CT.”

However, the data presented in Figure 2F/G/H do not show that it is the presence of signalling associated with activated fibroblasts (aFIB-RS), rather than fibroblasts alone or previously established TGFb/Fibroblasts signatures, that is the key determinant of these outcome correlations. Can the authors also present:

1) A GSEA plot of the aFib-RS comparing the Calon groups previously generated fibroblast v epithelial lineage data (GSE39396) to demonstrate if this is a de facto signature of fibroblasts. If this is highly correlated, then any use of this signature in bulk data will likely be identifying fibroblast-high tumours in general.

2) A survival plot and GSEA plots for their previously published F-TBRS using the same method to derive the best cut off; is the aFIB-RS better than this established method? The authors show that aFIB-RS and F-TBRS are highly correlated already; and while this new signature has been developed in a different way (oxali treatment to stimulate TGFb signalling) what is the benefit of this new signature beyond this and other fibroblast/stromal signatures?

3) A survival plot for their aFIB-RS in CMS4 tumours only; given that the authors propose that it is the signalling associated with activated stroma (rather than general CAF/stroma) that is the key here, if this new signature can identify the “bad stroma” tumours from the “not so bad stroma” tumours, then it should have different outcomes in CMS4?

4) Similar to above, can the aFIB-RS GSEA be plotted between relapse/non-relapse and responder/non-responder in CMS4 tumours only (appreciate these may be small numbers) Unless the aFIB-RS is decoupled from general TGFb/stromal signatures that already exist, it will remain an unknown if this new data that clearly has mechanistic interest provides any additional clinical/biomarker value. If this isn't shown, it remains unclear if a signature generated from fibroblasts v epithelium would be just as valuable as the aFIB-RS.

Point 2 - In line with the comments above, the value of POSTN presented in Figure 6 would also need to be performed in the same way, according to the stratification categories 1, 3, 4 above. Is POSTN identifying tumours with “bad stroma” rather than just identifying tumours with any stroma?

Point 3 - The conclusions from Figure 2 are quite strong given the data presented “our data indicate that fibroblasts are highly resistant to platinum-based drug compared to epithelial cancer cells”, considering this is based solely on 1 cell line in each lineage. Can the authors expand the number of cancer cell models used here?

Point 4 - Furthermore, in Figure 2D: Can the authors show that in this model if the oxali is still retained, and at what level, at the timepoints reached by the end of experiment, in the in vitro setting and at end of implants? This may help the discussion around whether the initial oxali treatment is all

that is needed to activate the fibroblasts to be more aggressive, or perhaps the retention of oxaliplatin is also essential.

Point 5 - Figure 3A-D: The authors show that fibroblasts display senescence markers after treatment with oxaliplatin, and also show clinical associations of these signatures with relapse/non-response; is it simply that these senescent signatures are elevated in fibroblasts compared to epithelium even in the absence of oxaliplatin, and therefore these signatures are surrogate markers for stromal content in general? Can the authors plot figures 3B&C with expression data from an epithelial cell model alongside to show this, or indeed to use the fibroblast v epithelial data from their GSE39396 dataset.

Point 6 - The right panel of Figure 3G indicates that this new signature is a strong surrogate for the previously defined Fib-TBRs, and figure 3H&I support this, although the Results and Discussion section doesn't really get across the new clinical value this new aFib-RS provides?

Point 7 - The focus on IL11 in Figure 4, which has a very weak R correlation value in the patient data should be explained more in the results, otherwise I would suggest moving this to the supplementary. When compared to the strength of the data in Figure 5, with high correlations and extreme differentials, alongside really eloquent and convincing TBRi data, the IL11 data seems underwhelming.

Useful if possible to include:

A) Can the authors show if oxaliplatin is taken up in normal stroma, perhaps in patient sample where this is a region of tissue separated from the tumour?

B) Figure 2E: Can this be presented in the same way as figure 2b/c?

C) The analyses in Figure 2 would benefit from a plot for aFIB-RS according to CMS (similar to Figure 6).

D) While the patients samples will likely have been exposed to 5FU+oxali therapies, can the authors include some of the analyses on their fibroblast line with 5FU+oxali, alongside oxali alone, as it is unlikely that oxaliplatin would even be given as a mono-therapy for CRC.

Typo: Line 166: do the authors mean Fig 3D?

Response to reviewers' comments

We thank the reviewers for their insightful comments that have improved this revised version to a large extent. You will find below a point-by-point reply detailing how we have addressed each of the reviewers concerns. We have labeled new data in blue in the text.

Reviewer #1 (Remarks to the Author): with expertise in colorectal cancer, cancer associated fibroblasts

Linares et al describe a mechanism for therapeutic resistance to chemotherapy in colon cancer, mediated by CAFs through accumulation of platinum-based drugs. They began their study by assessing platinum distribution in FFPE tumor samples from CRC patients, and found that oxaliplatin accumulated predominantly in the stroma, up to months following treatment, which was significantly longer than the accumulation in cancer cells. To mechanistically dissect this, they measured oxaliplatin accumulation in cancer cells and colon fibroblasts, in vitro, and found that fibroblasts were significantly more resistant to this treatment compared to cancer cells. They then coinjected fibroblasts with cancer cells into nude mice, and found that platinum-treated fibroblasts induced faster growing tumors compared to non-treated fibroblasts. They performed microarray-based transcriptional profiling to determine a gene signature associated with platinum-resistance in fibroblasts and used this signature to interrogate patient datasets. They found that expression of these genes is associated with poor survival and resistance to treatment. They performed additional bioinformatic analyses and chose to focus on TGF-beta and IL11. They found that IL11 expression was induced in colon fibroblasts as well as patient CAFs in response to platinum treatment and this could be inhibited by tgf-beta inhibitors. They overexpressed IL11 in cancer cells injected into nude mice and found that this increased tumor progression. Finally they show that POSTN expression in fibroblasts is also induced following platinum treatment and suggest that POSTN could be a potential biomarker for response to therapy.

Overall this is an interesting study, of a timely topic – the role of CAFs in mediating drug resistance. Nevertheless there are conceptual and technical limitations to this study that must be addressed to support the far-reaching conclusions made by the authors. In particular, this reviewer is not convinced that CAFs are the ones mediating resistance in patients (rather than other cells in the TME) or that this mechanism contributes to resistance in patients.

Specific comments:

1. Currently there is insufficient evidence for accumulation of platinum in CAFs rather than other stromal cells such as endothelial, immune etc. The authors show patient samples in Figure 1, but no costaining for CAF markers. Perhaps this is mediated by other cells in the TME? More images, more patients and most importantly costaining are needed to make this point convincing.

In order to address this referee's concerns regarding the impact of oxaliplatin on stromal cells other than CAFs, we assessed its immediate effect on the TME in an immunocompetent murine model of aggressive CRC grown from mouse tumor organoids (MTO) injected into the caecum wall of immunocompetent mice as described by Tauriello and colleagues¹. In more detail, MTOs were previously derived from primary tumors arising in genetically engineered mouse models with compound genetic alterations (Apc, Kras, Trp53, Tgfbr2)¹. For bioluminescence tracking, MTOs

were infected with a lentivirus encoding an eGFP–firefly luciferase fusion reporter construct under the control of the P_{gk1} promoter ². For culture expansion, MTOs were embedded in basement-membrane extract (BME) medium (Cultrex BME Type 2, Amsbio) and cultured at 37 °C with 85–90% humidity, atmospheric O₂ and 5% CO₂ in advanced DMEM/F12 supplemented with 10 mM HEPES, Glutamax, B-27 without retinoid acid (all Life Technologies), 100 ng/ml recombinant NOGGIN and 50 ng/ml recombinant EGF (Peprotech). 0.1 × 10⁶ cells in 70% BME were injected with a 30G syringe under binocular guidance into the submucosal wall of the distal caecum of 7-9 weeks old C57BL/6J mice purchased from Janvier Labs. Mice bearing tumors were randomly assigned to treatment (n=8) and control groups (n=5). Treated mice were injected IP with oxaliplatin (12 mg/kg) 96h and 24h before tumor resection.

Bioluminescence tracking *in vivo* showed a reduction in cancer cells abundance upon treatment (figure 1a). Following resection, tumor samples were analyzed by IHC to identify α-SMA (+) CAFs, CD31 (+) endothelial and CD45 (+) immune cells. We observed a significant reduction of blood vessels density and immune cells infiltration following therapy (figure 1a, supplementary figure 1a). In contrast, CAFs abundance remained unchanged (figure 1a, supplementary figure 1a) thus indicating that in contrast with cancer, endothelial and immune cells, CAFs are highly resistant to oxaliplatin.

Furthermore, we assessed oxaliplatin cytotoxicity *in vitro* against endothelial and immune cells in addition to fibroblasts and cancer cells. We observed that immune, endothelial and cancer cells were extremely sensitive to oxaliplatin compared to fibroblasts, thus corroborating above mentioned observation *in vivo*. Results are displayed in figure 1b,c and supplementary figure 1b-f. We performed LA-ICP-MS studies in additional patients' samples that confirmed the increased CT uptake in the stromal compartment compared to the cancer compartment of the tumor (figure 1f). Elemental bioimaging and histological evaluation by expert pathologist indicated that tumor areas displaying increased platinum uptake were enriched with fibroblasts (figure 1e, supplementary figure 2a). In addition, we performed IHC staining to identify the fibroblast population in tumors sections using FAP as a marker of cancer-associated fibroblasts ³. FAP (+) staining overlapped with areas showing increased platinum uptake (supplementary figure 2b). Collectively, our results indicate that in contrast with cancer, endothelial and immune cells, CAFs are highly resistant to oxaliplatin. Our data do not exclude the fact that endothelial or immune cells response to oxaliplatin may impact therapeutic outcome. Yet, these results suggest that endothelial and immune cells present at the time of treatment are cleared from the tumor, eventually being replaced over time by new cells recruited in the tumor after treatment. Therefore, we conclude that long term platinum uptake occurs mainly in the CAFs present at the time of treatment and resisting to chemotherapy. Main text and Methods were modified accordingly.

2. Following up on this, the authors treat fibroblasts in-vitro with platinum and show that the CM and/or cells induce resistance. Would this not have been achieved by treatment of any other cell type (again, endothelial and immune but also others - even cancer cells) with platinum?

Following this reviewer suggestion, we performed similar experiments with endothelial, immune and cancer cells. For this, we obtained condition media (CM) from each cell subtype treated with chemotherapy. As established in point 1, oxaliplatin displays strong cytotoxicity against these cell

types. Hence, each cell subtype was treated with its own oxaliplatin concentration EC_{50} in order to avoid complete cell death in this experimental setting. Results displayed in supplementary figure 3b indicate that CM derived from oxaliplatin-treated endothelial, immune and cancer cell did not induce measurable cancer cell resistance to treatment. Main text and Methods were modified accordingly.

3. Similarly, the authors must show that changes in TGF-beta and IL11 signaling indeed occur in CAFs and not in other cells in the TME (costaining in patients or mice). In fact they show the contrary – that modulation of IL11 in cancer cells created the same effect. It is possible that treating cancer cells with CM from epithelial or endothelial cells pretreated with platinum, would have yielded similar effects to the CAFs. Moreover, how do the authors exclude the effect of residual chemo in the medium?

In order to address this reviewer comment, we performed analyses on immune, cancer and endothelial cells treated with oxaliplatin in the experimental conditions previously set for cell types highly sensitive to oxaliplatin (Please refer to point 2 for further details). Results displayed in supplementary figure 4d show that *TGFB1* was not upregulated upon treatment in immune, cancer or endothelial cells. *IL11* was not detectable either in the presence or absence of oxaliplatin in these cell subtypes. With regard to the effect of residual chemotherapy in media, we agree with this reviewer that the experimental setting description deserves to be further clarified. Conditional media (CM) experiments were conducted as follow. After pre-stimulation, oxaliplatin-containing media were discarded and cell cultures were washed 3 times with HBSS (Lonza) to ensure the removal of residual free chemotherapy. Next, fresh DMEM supplemented with 0.2% FBS was added to the culture. CM was collected after 48 hours culture. Main text and Methods were modified accordingly.

4. Moreover, the sequencing data is from bulk, so once again these changes in gene expression may not be specific to CAFs. In fact the pathways most differentially expressed are indeed generic to drug treatment - P53 pathway, DNA repair and cellular apoptosis. The authors focus on TGF-beta, but changes in TGF-beta are not specific to drug treatment but rather a hallmark of the transition from normal fibroblasts to myofibroblastic CAFs. Certainly there are genes that are better hallmarks of SASP than TGF-beta.

Following this reviewer suggestion, we extended the analysis of SASP factors to *IL1b*⁴, *GDF15*⁵ and *CXCL8*⁴ in addition to TGF-beta 1⁶⁻⁸. Similar to *TGFB1*, we found that *IL1B*, *GDF15* and *CXCL8* expression levels were significantly increased upon treatment, thus suggesting SASP activation in fibroblasts treated with oxaliplatin. Results are displayed in supplementary figure 4c. With regard to the pathways most differentially expressed and generic to drug treatment, P53 pathway, DNA repair and apoptosis hallmarks were evaluated in monocultured fibroblasts treated with oxaliplatin compared to non-treated ones. These data are now displayed in supplementary table 2a. In addition, data showing SASP-S, SenRS and Fib-TBRS enrichment in monocultured fibroblasts treated with oxaliplatin are displayed in figure 3a,h. Main text was modified accordingly.

5. The data regarding POSTN is nice and looks like it is staining CAFs but again no costaining with CAF markers is shown. Also, how was the quantification done?

We have now performed IHC against FAP as a marker of cancer-associated fibroblasts³ and POSTN in consecutive section of CRC tumors. Analysis by expert pathologist indicated that POSTN is expressed by FAP (+) fibroblasts in the tumor. Results are displayed in supplementary figure 6c. POSTN staining analyses were performed using histological scoring (H-Score)⁹. In more detail, intensity scores (from 0 to 3) and intensity scores frequencies (from 0 to 100%) were assessed for each sample by expert pathologists. A final H-score representing stromal POSTN overall staining intensity was calculated as follow: $(\% \text{ of scored } 1 \text{ stromal area}) \times 1 + (\% \text{ of scored } 2 \text{ stromal area}) \times 2 + (\% \text{ of scored } 3 \text{ stromal area}) \times 3$. Main text and Methods were modified accordingly.

6. The authors mention the use of combination treatment today rather than platinum alone which brings the question of relevance of these findings, especially since the mouse study was done in immune-deficient mice, thus already limiting the TME responses.

Indeed, current standard-of-care involves either fluoropyrimidines (5-fluorouracil, capecitabine) in early stage CRC or their combination with oxaliplatin in later stages. However, as demonstrated by the results of the MOSAIC trial¹⁰ and the works referenced in the introduction^{11,12}, oxaliplatin administration offers little benefit to fluoropyrimidines, and a high proportion of patients will still relapse. Yet, the addition of oxaliplatin to fluoropyrimidines regimen remains the main treatment option in aggressive CRC^{10,13,14}. This problem led us to use oxaliplatin alone as a system to understand the influence this drug exerts during tumor relapses. Strengthening our concern, we saw that a major proportion of administered platinum, as seen in figure 1, was retained in the tumor stroma. Given the scope of the present manuscript, the use of immunodeficient mice allowed us both to focus on particular CAF-derived responses without the influence of immune responses while accommodating human-derived cellular models. While we would not discourage the administration of oxaliplatin to CRC patients, the relevance of our results points to the clinical use of the genetic signatures from platinum-activated fibroblasts to predict the actual benefit of oxaliplatin. In addition, our data suggest that POSTN could be used as a biomarker for clinical diagnosis using paraffin-embedded formalin-fixed tumor samples to assist clinical decision-making. Main text was modified accordingly.

Minor comments:

1. The authors should describe in the intro the CRC subtypes (row 65)
2. Many of the graphs show bars and not dots and it is hard to assess how the replicates distribute.
3. Figure 2F – how was high vs low determined? It is not the median, so what was the criterion?

Main text was modified according to this reviewer suggestion.

[...]For these reasons, elucidating the determinants of platinum-based treatment effectiveness is paramount to improve the standard of care for cancer patients. Much effort has been made to decipher the mechanisms of platinum resistance intrinsic to epithelial cancer cells [1,6] and

multiple molecular classifications were developed in the recent years, enabling the identification of CRC subtypes ~~C-type, Stem-like, CCS3 or CMS4~~ associated with worse patient's outcome [7–10] and lack of benefit from CT, including oxaliplatin [10,11]. Initially, there was not an apparent molecular consensus between the poor prognostic groups of CRC, which were either enriched in Wnt signaling and markers associated with cancer stem cells (Stem-like)¹⁵, in serrated tumors (CCS3)¹⁶, or in mesenchymal tumors with deficient mismatch repair machinery (C-type)¹⁷. In an effort to integrate the previously published classification algorithms, Guinney and colleagues defined four Consensus Molecular Subtypes (CMS) of CRC, namely CMS1 (MSI immune), CMS2 (canonical), CMS3 (metabolic) and CMS4 (mesenchymal), from which CMS4 tumors showed the worse clinical outcome¹⁸. However, these classification systems and advances in the genetics of cancer have not yet translated into efficient treatment synergizing with platinum drugs and failed so far to provide biomarkers of therapeutic response [12,13].

All histograms are now displayed with dots corresponding to experimental replicates.

To accommodate reviewer 4 suggestions, optimal cutoff was defined as the value giving the most significant split calculated by log-rank test using `surv_cutpoint`, `survfit` and `coxph` functions as implemented in Survival and Survminer R packages. Methods were modified accordingly. Figure 2F is now displayed as figure 2c in this revised manuscript.

Reviewer #2 (Remarks to the Author): with expertise in colorectal cancer, periostin

In this manuscript, Linares et al found that platinum-based drugs accumulate in cancer-associated fibroblasts (CAFs) long time after treatment in colorectal cancer (CRC) patients. Oxaliplatin accumulation activates the TGF- β pathway in CAFs, which induces the expression of IL-11 and periostin (POSTN). POSTN isoform 4 is the main splicing variant upregulated in CAFs and is a marker of poor prognosis. The pro-oncogenic secretory phenotype in CRC fibroblasts induced by oxaliplatin accumulation promotes CRC progression and resistance to chemotherapy. The authors discover an interesting and important finding that platinum accumulation in CAFs promotes CRC progression, which may contribute to CRC treatment. However, there is lots of correlation analyses in this manuscript but the directly regulation mechanism is not very clear. Multiple other concerns are needed to be addressed:

Major points:

1. The authors used nude mice subcutaneously injected CRC cells and CAFs to do the in vivo studies. It is better to add a spontaneous CRC mouse model (APC mim/+) or AOM/DSS-induced CRC mouse model, which can receive the treatment with platinum-based drugs to detect the activation of TGF- β , upregulation of IL-11 and POSTN in CAFs and the related regulation mechanism.

Following this reviewer suggestion, we sought to perform oxaliplatin treatment in a murine model of orthotopic aggressive CRC. Unfortunately, APC+/min and AOM/DSS models give rise to tumors lacking mutations in the key driver genes involved in colon cancer. Accordingly, these tumors develop as rather benign, non-invasive/non-aggressive polyps with very limited stroma/CAFs recruitment¹⁹. In addition, AOM/DSS is a model of inflammatory tumors that account for only 1-

2% of human colon tumors while APCmin/+ tumors are adenomas arising in the small intestine and not in the colon¹⁹. Of note, patients presenting similar pathology are generally subjected to colectomy and not to platinum-based chemotherapy²⁰. We thus decided to take advantage of a model of aggressive CRC grown from mouse tumor organoids (MTO) injected into the caecum wall of immunocompetent mice as described by Tauriello and colleagues¹. In more details, MTOs were previously derived from primary tumors arising in genetically engineered mouse models with compound genetic alterations (Apc, Kras, Trp53, Tgfbr2)¹. For bioluminescence tracking, MTOs were infected with a lentivirus encoding an eGFP–firefly luciferase fusion reporter construct under the control of the P_{gk1} promoter². For culture expansion, MTOs were embedded in basement-membrane extract (BME) medium (Cultrex BME Type 2, Amsbio) and cultured at 37 °C with 85–90% humidity, atmospheric O₂ and 5% CO₂ in advanced DMEM/F12 supplemented with 10 mM HEPES, Glutamax, B-27 without retinoid acid (all Life Technologies), 100 ng/ml recombinant NOGGIN and 50 ng/ml recombinant EGF (Peprotech). 0.1 × 10⁶ cells in 70% BME were injected with a 30G syringe under binocular guidance into the submucosal wall of the distal caecum of 7-9 weeks old C57BL/6J mice purchased from Janvier Labs. Mice bearing tumors were randomly assigned to treatment (n=8) and control groups (n=5). Treated mice were injected IP with oxaliplatin (12 mg/kg) 96h and 24h before tumor resection.

Bioluminescence tracking *in vivo* showed a reduction in cancer cells abundance upon treatment (figure 1a). Following resection, tumor samples were analyzed by IHC to identify α-SMA (+) CAFs, CD31 (+) endothelial and CD45 (+) immune cells. We observed a significant reduction of blood vessels density and immune cells infiltration following therapy (figure 1a, supplementary figure 1a). In contrast, CAFs abundance remained unchanged (figure 1a, supplementary figure 1a) thus indicating that in contrast with cancer, endothelial and immune cells, CAFs are highly resistant to oxaliplatin. Furthermore, IHC analyses showed increased P-SMAD3 (supplementary figure 4f,g) and P-STAT3 nuclear staining after therapy (supplementary figure 5a), thus indicating the activation of TGF-β and IL11 related signaling. POSTN expression was also significantly upregulated in mice tumors upon oxaliplatin treatment (figure 5d). Main text and Methods were modified accordingly.

2. Based on the authors' results, oxaliplatin combined with POSTN blockade supposes to achieve better therapeutic efficacy for those patients unresponsive to oxaliplatin chemotherapy. It will be very interesting to design animal studies to verify the authors' hypothesis.

We fully agree with this reviewer, the results of this study suggest that strategies blocking POSTN4 expression may restore response to treatment in unresponsive patients. Further investigation may lead to the development of specific inhibitors of POSTN expression. Main text was modified accordingly.

3. Fig. 2d show the Kaplan-Meier plot of nude mice injected subcutaneously with CRC cells alone, co-injected with CCD18co non-treated or pre-treated with oxaliplatin. What about tumor size? Does CCD18co pre-treated with oxaliplatin influence tumor growth and tumor cell proliferation?

Figure 2d is now displayed as figure 2a in the revised version of the manuscript. To address this reviewer concern, we followed tumor expansion after initiation in mice co-injected with either CRC cells plus untreated CCD-18co or CRC cells plus CCD-18co pre-treated with oxaliplatin. Data

shown in supplementary figure 3a indicate that after initiation, tumor growth followed a similar trend in both conditions. These findings suggest that pre-treated fibroblasts co-injected with CRC cells promote tumor initiation rather than tumor expansion. This result may be explained by the transitory impact of co-injected fibroblasts on the tumor development. Indeed, murine stroma is actively recruited during tumor expansion and CAFs of murine origin are replacing co-injected fibroblasts over time³. Main text was modified accordingly.

4. Based on Fig. 3h the Fib-TBRS was significantly increased in oxaliplatin-treated CCD-18co, the authors concluded that “an autocrine activation of the TGF-beta pathway in fibroblasts” (Line 179). Previous evidence show that TGF-beta is mainly produced by colorectal tumor cells in CRC. Therefore, representative protein levels (TGF- β and/or p-Smad3, p-Smad4) need to be detected in CAFs and in tumor tissues.

Stromal activation of the TGF-beta pathway clearly associates with worse patient outcome^{21,22} and several lines of evidence indicate that CAFs are the main source of TGF-beta in CRC²³⁻²⁵. For instance, cell sorting by flow cytometry of CAFs, cancer cells, immune cells and endothelial cells followed by transcriptomic analysis indicated that *TGFB* levels were significantly higher in CAFs compared to CRC cells^{1,21}. Here, we performed IHC analysis of P-SMAD3 in tumors from mice treated with oxaliplatin. P-SMAD3 nuclear staining was increased in stromal cells after therapy (supplementary figure 4f,g), thus indicating the activation of TGF-beta signaling (please refer to point 1 for further details). In addition, we performed P-SMAD3 detection by western blot in cultured fibroblasts treated with oxaliplatin. Results displayed in supplementary figure 4h indicate that P-SMAD3 is increased upon treatment in fibroblasts. Main text was modified accordingly.

5. Fig. 4b-e shows the transcription levels of IL-11 upregulated after oxaliplatin treatment, and the protein levels of IL-11 should also be tested.

7. Fig. 5c-d shows the transcription levels of POSTN upregulated in CAFs after oxaliplatin treatment, and the protein levels of POSTN should also be tested.

In order to address reviewer comments 5 and 7, protein levels of POSTN and IL11 were measured in cultured fibroblasts treated with oxaliplatin. Results displayed in supplementary figure 5b and 6d respectively indicate that IL11 and POSTN protein levels are increased in fibroblasts upon treatment. Main text was modified accordingly.

6. In Fig. 4e TGF- β pathway inhibitor partially reduces the expression levels of IL-11 induced by oxaliplatin treatment. What other signaling pathways other than the TGF- β pathway also regulate IL-11 expression after oxaliplatin treatment? In addition, how does oxaliplatin accumulation activate the TGF- β pathway in CAFs?

8. In Fig. 5f TGF- β pathway inhibitor partially reduces the expression levels of POSTN induced by oxaliplatin treatment. Apart from the TGF- β pathway, is POSTN expression regulated by other pathways after oxaliplatin treatment?

Increasing evidence suggest that senescent cells are generated by a broad range of systemic anti-cancer chemotherapies, and can potentially fuel many aspects of cancer progression²⁶. Our data

suggest that oxaliplatin incorporation induces a senescent state in CAFs, which leads to an enhanced senescence-associated secretory phenotype (SASP). Of note and as previously reported by multiple works, TGF-beta is one of the main components of the SASP ²⁷⁻²⁹. Therefore, we believe that oxaliplatin-induced senescence in CAFs leads to an increased TGF-beta production, which in turn activates the TGF-beta pathway in an autocrine manner ^{3,25,30}.

On the other hand, we could not find any report pointing to additional pathways other than TGF-beta that could regulate IL11 or POSTN after oxaliplatin treatment. Interestingly, IL11 is a factor secreted by senescent fibroblasts in lung ³¹. Although in non-treatment conditions, the expression of both IL11 and POSTN could be mediated by other signaling pathways such as oncogenic Ras for IL-11 ³² or IL6 and Notch for POSTN ^{33,34}, TGF-beta remains the main inducer of IL11 and POSTN expression in CAFs ^{3,35-42}. The gene expression profile of oxaliplatin treated fibroblasts produced for this study (GSE181020) didn't suggest any modulation of IL6 expression. In our hands, increased expression of IL6 did not result in POSTN accumulation in the tumor. Given the wide range of possibilities, we cannot be certain about the relevance of pathways that might be involved, and we believe that assessing this question would be a subject for a much broader study. Nevertheless, our data show a strong effect in downregulating *IL11* (figure 4e), *POSTN* (figure 5b,f,g) and *POSTNi4* expression (figure 7c) when TGF-beta pathway inhibitors are used, which confirms the major role of TGF-beta in regulating IL11 and POSTN. Main text was modified accordingly.

Minor points:

1. Fig. 7f and Fig. 7g are the same experiments of different groups. Combine Fig. 7f and Fig. 7g into one graph.

Following this reviewer's suggestion, figure 7f and 7g are now displayed as figure 7f.

Reviewer #3 (Remarks to the Author): with expertise in colorectal cancer, gene signatures

CMS4 subtype colorectal cancer is characterized by poor clinical outcome even after standard chemotherapy with 5-fluorouracil/leucovorin with or without oxaliplatin. In a retrospective analysis of NSABP clinical trial C-07 which evaluated the efficacy of oxaliplatin added to 5-FU/leucovorin, patients with CMS4 subtype colon cancer did not benefit from the addition of oxaliplatin. CMS4 tumors are characterized by high stromal content and resulting gene expression signature mimicking epithelial mesenchymal transition – however, investigators have demonstrated that EMT and TGF-beta signature of CMS4 tumors originate from stromal fibroblasts rather than cancer cells, although conflicting data do exist. Mechanism of oxaliplatin resistance of CMS4 colon cancer remained unknown.

Linares et al attempted to identify the mechanism of oxaliplatin resistance of CMS4 tumors. First, they checked the presence of oxaliplatin in the tumor tissue after therapy in 31 residual colorectal cancer. This experiment demonstrated a presence of residual oxaliplatin only the stromal cells mainly cancer associated fibroblasts (CAF). In vitro experiments were conducted using CAFs to demonstrated that TGF-beta signaling was augmented by the retained oxaliplatin resulting in secretion of multiple factors associated with poor prognosis, such as IL11, a prometastatic cytokine. Periostin (POSTN), especially a specific isoform POSTNi4, was identified

as a marker that is induced by oxaliplatin treated CAFs and forced expression of POSTN^{is4} in colon cancer organoids resulted in oxaliplatin resistance. POSTN overexpression in the tumor stroma was associated with poor clinical outcome and was associated with CMS4 subtype. The study successfully identified the mechanism of oxaliplatin resistance in CMS4 colon cancer and suggest potential ways of overcoming chemo-resistance through intervention of molecules such as IL11 and POSTN secreted by CAFs.

Currently there is no widely used clinical test to identify CMS4 colon cancer subtype. This study identifies POSTN immunohistochemistry as a potential clinical surrogate marker of CMS4 subtype.

Soonmyung Paik, MD
Professor
Yonsei University College of Medicine

We thank this reviewer for its positive evaluation of the manuscript.

Reviewer #4 (Remarks to the Author): with expertise in colorectal cancer, gene signatures, CRC stroma

In this study by Linares and colleagues, the authors build on new elemental imaging observations of oxaliplatin accumulation in the stromal components of tumour tissue, to develop and characterise oxaliplatin activated fibroblasts. Molecular signatures induced in these activated fibroblasts are aligned to clinical correlates, where they are associated with poor outcomes and limited response to therapy. The authors eloquently demonstrate the essential nature of TGF β and POSTN signalling in this molecular axis.

The authors have presented an interesting and important piece of work, and have utilised existing models and molecular data to demonstrate the associations with the transcriptional targets upregulated in oxaliplatin activated stroma. These new data fit with the group's prior publications into the role and importance of the stroma and TGF β signalling in colorectal cancer, and will make a valuable contribution to the field. The finding of oxaliplatin accumulation is certainly of mechanistic importance, and exploitation of this will inevitably have therapeutic relevance. There is limited mechanistic data presented as to how this accumulation happens, or how it can either be prevented or reduced.

The main data in the paper looks to utilise the signature generated from oxaliplatin treated fibroblasts, which display increased TGF β signalling, therefore my main question is to what extent that this new signature provides more useful information above the numerous stromal signatures that exist, and the TGF β -stimulated fibroblast signatures previously developed/tested by the authors. I have suggested a number of points below that I hope will give the reader additional confidence in the data presented, and have also asked for some specific analyses that it is hoped will enable the authors to demonstrate if this molecular insight oxaliplatin-activated fibroblast signalling provides additional biomarker or mechanistic value beyond the groups previously defined fibroblast signatures.

Main essential points:

Point 1 - The authors state that, in line 146-148: "Gene set enrichment analysis (GSEA) showed that aFib-RS was significantly upregulated in tumors from unresponsive (Figure 2g) and relapsing patients (Figure 2h), thus indicating an association between the gene program activated in fibroblasts upon oxaliplatin treatment and the lack of benefit from CT." However, the data presented in Figure 2F/G/H do not show that it is the presence of signalling associated with activated fibroblasts (aFIB-RS), rather than fibroblasts alone or previously established TGFb/Fibroblasts signatures, that is the key determinant of these outcome correlations. Can the authors also present:

1) A GSEA plot of the aFib-RS comparing the Calon groups previously generated fibroblast v epithelial lineage data (GSE39396) to demonstrate if this is a de facto signature of fibroblasts. If this is highly correlated, then any use of this signature in bulk data will likely be identifying fibroblast-high tumours in general.

We do agree with this reviewer's concern, the transcriptomic analysis of bulk RNA from whole tumor has several limitations. For instance, signatures are generally developed based on the genetic profile of a cell type of interest upon stimulation. Hence, the enrichment of derived signatures may probably inform about both the abundance of this cell subtype and the activation of a specific gene program. As suggested by this reviewer, we applied GSEA to the analysis of aFib-RS in cancer cells and CAFs. For this, the transcriptomic gene profiles of EpCAM+ and FAP+ FACS purified cell subsets were retrieved from previous studies (GSE39396). GSEA showed that aFib-RS was significantly enriched in CAFs compared to cancer cells, thus indicating that genes in the aFib-RS are preferentially expressed by CAFs (supplementary figure 3c). In order to decouple the confounding effect of CAFs hallmarks in aFib-RS, we identified the aFib-RS probset specifically expressed by CAFs compared to all other cell subtypes in GSE39396 dataset ($FC > 1$; $P < 0.05$; $FDR < 0.01$). We tested the value of CAF specific and CAF non-specific aFib-RS (FAP (+) aFib-RS and FAP (-) aFib-RS respectively) to segregate relapsing and non-relapsing patients (in GSE17536 and GSE39582; supplementary table 1b) as well as patients responsive and unresponsive to chemotherapy (in GSE14333 and GSE72970; supplementary table 1c). In all cases and similar to the full aFib-RS signature, the two aFib-RS subsets were enriched in relapsing patients and tumors resistant to treatment (Figure 2c,d,g,h, supplementary table 1b,c). Collectively, these results do not exclude the intrinsic weight of the cell subtype. Yet, they suggest that the predictive power of aFib-RS may not only rely on CAF hallmarks. Main text and Methods were modified accordingly.

2) A survival plot and GSEA plots for their previously published F-TBRS using the same method to derive the best cut off; is the aFIB-RS better than this established method? The authors show that aFIB-RS and F-TBRS are highly correlated already; and while this new signature has been developed in a different way (oxali treatment to stimulate TGFb signalling) what is the benefit of this new signature beyond this and other fibroblast/stromal signatures?

For this study, we used GSE181026 dataset to derive the Fib-TBRS. The main difference with previous F-TBRS²¹ lies on the TGF-beta treatment performed in media containing very low (0.2%)

FBS concentration. Compared to F-TBRS derived from cells treated in media containing 2% serum, Fib-TBRS shows increased amount of genes differentially regulated by TGF-beta.

To review GEO accession GSE181026:

Go to <https://www.ncbi.nlm.nih.gov/geo/query/acc.cgi?acc=GSE181026>

Enter token oloxwuedtupah into the box

GSEA analyses were performed for aFib-RS in figure 2g,h and for Fib-TBRS in figure 3i. Normalized enrichment score was slightly higher for Fib-TBRS compared to aFib-RS in the GSE14333 subset (NES=1.61 and NES=1.34 respectively). In contrast, normalized enrichment score was higher for aFib-RS compared to Fib-TBRS in the GSE72970 subset (NES=1.92 and NES=1.51 respectively).

We next categorized the cohort from GSE17536 according to high or low Fib-TBRS and high or low aFib-RS expression using a cutoff defined as the value giving the most significant split calculated by log-rank test using `surv_cutpoint`, `survfit` and `coxph` functions as implemented in `Survival` and `Survminer` R packages. As expected, high expression of Fib-TBRS as well as FAP (+) and (-) Fib-TBRS subsets were associated with poor prognosis (Supplementary Table 2c). In this dataset, while association of both aFib-RS and Fib-TBRS predicted poor prognosis in a similar fashion, Fib-TBRS showed a higher Hazard Ratio (HR) than that of aFib-RS (3.47 vs 2.49 respectively) (Figure 2c, Supplementary Table 2c). To validate these observations, and to answer the following questions from this reviewer, we used the GSE39582 dataset, which includes 519 CRC patients with clinical outcome data. Again, elevated expression of Fib-TBRS and FAP (+) or (-) corresponding subsets were associated with poor prognosis (supplementary table 2c). In this case, the HR for Fib-TBRS was still higher than for aFib-RS, although the difference was severely diminished (1.73 vs 1.62 respectively; figure 2d, supplementary table 2c). In addition and as detailed in the following point 1-3, only aFib-RS segregated relapsing patients in the CMS4 subset (figure 2f, supplementary table 2c). Since they hold a similar predictive power, and given their correlation represented in figure 3g, we believe that both aFib-RS and Fib-TBRS genetic signatures could be used to determine tumors with poor outcome according to their stromal composition. However, as discussed in point 1-3, the predictive power of aFib-RS is superior to that of Fib-TBRS in stromal CRC tumors (CMS4), which suggests that expression of aFib-RS can further identify CRC with tumor promoting/protective stroma. Main text and Methods were modified accordingly.

3) A survival plot for their aFIB-RS in CMS4 tumours only; given that the authors propose that it is the signalling associated with activated stroma (rather than general CAF/stroma) that is the key here, if this new signature can identify the “bad stroma” tumours from the “not so bad stroma” tumours, then it should have different outcomes in CMS4?

To answer this question, we first classified all patients from GSE39582 using the Random Forest method from `CMSclassifier` package in R¹⁸, and selected those patients with a CMS4 probability above 0.5 to discard ambiguous results. A total of 121 samples were classified as CMS4, which was sufficient to produce statistically relevant observations. Interestingly, increased expression of aFIB-RS and FAP (+) or (-) related subsets significantly predicted poor outcome in these patients (figure 2f, supplementary table 1b). In contrast and as mentioned in point 1-2, Fib-TBRS did not provide statistically relevant segregation in CMS4 (supplementary table 2c). This could be explained by the fact that CMS4, which includes tumors with elevated expression of stromal and TGF-beta signatures¹⁸, cannot be further subdivided by Fib-TBRS. Therefore, in agreement with the

hypothesis raised by this reviewer, aFIB-RS levels segregate the stromal tumors with better or worse prognosis. Main text and Methods were modified accordingly.

4) Similar to above, can the aFIB-RS GSEA be plotted between relapse/non-relapse and responder/non-responder in CMS4 tumours only (appreciate these may be small numbers) Unless the aFIB-RS is decoupled from general TGFb/stromal signatures that already exist, it will remain an unknown if this new data that clearly has mechanistic interest provides any additional clinical/biomarker value. If this isn't shown, it remains unclear if a signature generated from fibroblasts v epithelium would be just as valuable as the aFIB-RS.

As for the study of GSE39582 detailed above, we classified patients from either GSE14333 or GSE72970 subsets using the Random Forest method from CMSclassifier package in R. We were able to identify only 4 CMS4 patients in GSE72970 (3 responsive / 1 unresponsive) and 10 in GSE14333 (7 non-relapsing / 3 relapsing respectively). We applied anyway GSEA to the GSE14333 subset. While not reaching statistical significance, an upregulation trend was observed in relapsing patients for aFib-RS (NES=1.19; supplementary figure 3d) but not for Fib-TBRS (NES=0.79; supplementary table 2d), thus suggesting that aFIB-RS may be able to identify stromal tumors with a higher degree of resistance to CT. Main text and Methods were modified accordingly.

Point 2 - In line with the comments above, the value of POSTN presented in Figure 6 would also need to be performed in the same way, according to the stratification categories 1, 3, 4 above. Is POSTN identifying tumours with "bad stroma" rather than just identifying tumours with any stroma?

As suggested by this reviewer, we analyzed the expression of *POSTN* in GSE39396. Our data showed that *POSTN* was specifically expressed by CAFs compared to CRC cells (supplementary figure 6b). In addition, we performed IHC against FAP as a marker of cancer-associated fibroblasts³ and *POSTN* in consecutive section of CRC tumors. Analysis by expert pathologist indicated that *POSTN* is expressed by FAP (+) fibroblasts in the tumor. Results are displayed in supplementary figure 6c.

While the reduced number of CMS4 cases in GSE14333 and GSE72970 did not allow to perform relevant analyses, data obtained from CMS4 samples in GSE39582 (please refer to point 1-3 for further details) indicated that *POSTN* expression was also associated with poor prognosis in stromal tumors (supplementary figure 7e).

In order to address this reviewer's more general concern regarding the identification of tumors with either "bad stroma" or tumors with any stroma, we performed a new set of analyses based on stained FFPE CRC sections. In contrast with whole-tumor transcriptomics which is limited by gene expression variations arising from changes in cancer cells abundance and stromal infiltration^{43,44}, FFPE staining enables the analysis of the stromal and cancer compartments in patient tumors. We compared the impact of stromal abundance and *POSTN* expression on cancer progression. For this, expert pathologists assessed the percentage of stroma (H&E stained sections) and *POSTN* expression (anti-*POSTN* Ac IHC stained sections) in FFPE tumor samples from the 109 CRC patients previously described in this study. For the record, *POSTN* expression level measurements were

performed using histological scoring (H-Score)⁹. In more detail, intensity scores (from 0 to 3) and intensity scores frequencies (from 0 to 100%) were assessed in each tumor sample and an H-score representing stromal POSTN overall staining intensity was calculated as follow: $(\% \text{ of scored } 1 \text{ stromal area}) \times 1 + (\% \text{ of scored } 2 \text{ stromal area}) \times 2 + (\% \text{ of scored } 3 \text{ stromal area}) \times 3$. This scoring method provides an expression intensity value independent of the extent of the desmoplastic reaction in the tumor.

Our results indicated that, similar to POSTN levels (figure 6a), stroma was more abundant in CMS4 tumors (figure 6b). However, stromal abundance did not correlate with POSTN staining intensity (supplementary figure 7f). Furthermore, and in contrast with the high POSTN expression that associates with worse prognosis (figure 6c), increased stromal abundance failed to segregate patients with higher risk of relapse (figure 6d). We next reproduced these analyses in the subset of CMS4 cases with follow-up data composed of 39 patients. Here again, stromal abundance did not correlate with POSTN expression (supplementary figure 7g) and increased stromal abundance didn't associate with higher risk of relapse (supplementary figure 7h). Yet, low POSTN intensity segregated a small subset of patients in CMS4 with no observed recurrences (supplementary figure 7i). Although it did not reach statistical significance, this result corroborates the transcriptomic data that associate increased *POSTN* levels with reduced survival in CMS4 patients (supplementary figure 7e). Prompted by these findings, we evaluated the prognosis power of POSTN in tumors with low or high stromal content. Remarkably, increased POSTN levels associated with worse outcome in patients with tumors displaying either high (>30%; figure 6e) or low (<30%; figure 6f) stromal load. Collectively, these data suggest that POSTN prognosis value is a reflection of a stromal activation status rather than stromal abundance in CRC tumors. Main text and Methods were modified accordingly.

Point 3 - The conclusions from Figure 2 are quite strong given the data presented “our data indicate that fibroblasts are highly resistant to platinum-based drug compared to epithelial cancer cells”, considering this is based solely on 1 cell line in each lineage. Can the authors expand the number of cancer cell models used here?

Following this reviewer recommendation, we have expanded the number of cells analyzed upon treatment. We assessed oxaliplatin cytotoxicity against immune (PBMC) and endothelial cells (HUVEC) in addition to fibroblasts (CCD-18co, CAF1, CAF2, CAF3) and cancer cells (HT29-M6, PDO, PDO2). We observed that cancer (figure 1b, supplementary figure 1b), immune (supplementary figure 1c) and endothelial cells (supplementary figure 1d) were extremely sensitive to oxaliplatin compared to fibroblasts (figure 1b, supplementary figure 1e). We next treated cultured CCD-18co, CAF1 and CAF2 as well as HT29-M6 cells, PDO and PDO2 with oxaliplatin for 12 days. Cancer cells did not survive to 9 days of treatment (figure 1c, supplementary figure 1f). In contrast, about 50-80% of fibroblasts resisted to up to 12 days of CT (figure 1c, supplementary figure 1f).

In addition, we assessed oxaliplatin early effect on the TME in a model of aggressive CRC grown from mouse tumor organoids (MTO) injected into the caecum wall of immunocompetent mice as described by Tauriello and colleagues¹. In more details, MTOs were previously derived from primary tumors arising in genetically engineered mouse models with compound genetic alterations (*Apc*, *Kras*, *Trp53*, *Tgfbr2*)¹. For bioluminescence tracking, MTOs were infected with a lentivirus encoding an eGFP–firefly luciferase fusion reporter construct under the control of the

Pgk1 promoter ². For culture expansion, MTOs were embedded in basement-membrane extract (BME) medium (Cultrex BME Type 2, Amsbio) and cultured at 37 °C with 85–90% humidity, atmospheric O₂ and 5% CO₂ in advanced DMEM/F12 supplemented with 10 mM HEPES, Glutamax, B-27 without retinoid acid (all Life Technologies), 100 ng/ml recombinant NOGGIN and 50 ng/ml recombinant EGF (Peprotech). 0.1 × 10⁶ cells in 70% BME were injected with a 30G syringe under binocular guidance into the submucosal wall of the distal caecum of 7-9 weeks old C57BL/6J mice purchased from Janvier Labs. Mice bearing tumors were randomly assigned to treatment (n=8) and control groups (n=5). Treated mice were injected IP with oxaliplatin (12 mg/kg) 96h and 24h before tumor resection.

Bioluminescence tracking *in vivo* showed a reduction in cancer cells abundance upon treatment (figure 1a). Following resection, tumor samples were analyzed by IHC to identify α-SMA (+) CAFs, CD31 (+) endothelial and CD45 (+) immune cells. We observed a significant reduction of blood vessels density and immune cells infiltration following therapy (figure 1a, supplementary figure 1a). In contrast, CAFs abundance remained unchanged (figure 1a, supplementary figure 1a) thus corroborating *in vitro* observations (figure 1b,c, supplementary figure 1b-f) and indicating that in contrast with cancer, endothelial and immune cells, CAFs are highly resistant to oxaliplatin. Main text and Methods were modified accordingly.

Point 4 - Furthermore, in Figure 2D: Can the authors show that in this model if the oxali is still retained, and at what level, at the timepoints reached by the end of experiment, in the *in vitro* setting and at end of implants? This may help the discussion around whether the initial oxali treatment is all that is needed to activate the fibroblasts to be more aggressive, or perhaps the retention of oxaliplatin is also essential.

Figure 2d is now displayed as figure 2a in this revised version of the manuscript. We followed tumor expansion after initiation in mice injected with either CRC cells plus untreated CCD-18co or CRC cells plus CCD-18co pre-treated with oxaliplatin. Data shown in supplementary figure 3a indicate that after initiation, tumor growth followed a similar trend in both conditions. These findings suggest that pre-treated fibroblasts co-injected with CRC cells promote tumor initiation rather than tumor expansion. This result may be explained by the transitory impact of co-injected fibroblasts on the tumor development. Indeed, mouse stroma is actively recruited during tumor expansion and CAFs of murine origin are replacing co-injected fibroblasts over time ³.

We performed ICP-MS in CCD-18co cultured *in vitro* a long time after oxaliplatin retrieval. We observed that platinum was still detectable in cultured CCD-18co at least 90 days after the end of the treatment (figure 1d). This suggest that long term platinum uptake in fibroblasts occurs also *in vitro* and may impact gene expression long after the end of the treatment. Indeed, *TGFB1* (supplementary figure 4e), *IL11* (figure 4d), *POSTN* (figure 5e) and *POSTNi4* (figure 7d) upregulation remained detectable in fibroblasts at least 90 days after oxaliplatin retrieval while following a decreasing trend over time paralleling the one observed for platinum uptake. These results suggest an association between platinum uptake levels and fibroblasts activation. Main text was modified accordingly.

Point 5 - Figure 3A-D: The authors show that fibroblasts display senescence markers after treatment with oxaliplatin, and also show clinical associations of these signatures with

relapse/non-response; is it simply that these senescent signatures are elevated in fibroblasts compared to epithelium even in the absence of oxaliplatin, and therefore these signatures are surrogate markers for stromal content in general? Can the authors plot figures 3B&C with expression data from an epithelial cell model alongside to show this, or indeed to use the fibroblast v epithelial data from their GSE39396 dataset.

As suggested by this reviewer, we conducted GSEA in CAFs and cancer cells from GSE39396 similar to the ones provided in point 1-1. SASP-S was enriched in CAFs compared to cancer cells, thus indicating that genes in the SASP-S are preferentially expressed by CAFs (supplementary table 2b). We analyzed the expression levels of the senescence markers displayed in figure 3b (*CDKN1A* and *CDKN2A*) and additional ones studied in this revised version of the manuscript (*IL1B*, *CXCL8* and *GDF15*) in GSE39396. *CXCL8* and *CDKN1A* were significantly upregulated in CAFs compared to cancer cells (supplementary figure 4b). In contrast, *IL1B*, *GDF15* and *CDKN2A* were either enriched in cancer cells compared to CAFs or were equally expressed by both cell subtypes (supplementary figure 4b). Similar to aFib-RS analyses (please refer to point 1-1) and in order to decouple the confounding effect of CAFs hallmarks in SASP-S, we identified the FAP (+) and (-) SASP-S probe sets in GSE39396 dataset. We tested the value of CAF specific and CAF non-specific SASP-S to segregate relapsing and non-relapsing patients (in GSE17536, GSE39582 and in the CMS4 subset from GSE39582; supplementary table 2c) as well as patients responsive and unresponsive to chemotherapy (in GSE14333 and GSE72970; supplementary table 2d). In all cases, FAP (-) SASP-S was enriched in relapsing patients and tumors resistant to treatment (supplementary table 2c,d). In contrast, FAP (+) SASP-S failed to identify disease recurrence in the CMS4 subset (GSE39582-CMS4; supplementary table 2c) and relapsing patients after therapy (GSE14333; supplementary table 2d). Collectively, these results do not exclude the intrinsic weight of the cell subtype. Yet, they suggest that the predictive power of SASP-S may not exclusively rely on CAF hallmarks. Main text and Methods were modified accordingly.

Point 6 - The right panel of Figure 3G indicates that this new signature is a strong surrogate for the previously defined Fib-TBRS, and figure 3H&I support this, although the Results and Discussion section doesn't really get across the new clinical value this new aFib-RS provides?

Results were updated and discussion section is now modified as follow to accommodate points 6 and 7 from this reviewer while taking into consideration the answers to previous points:

Line 412

...both aFib-RS and Fib-TBRS described in this study could be used to determine tumors with poor outcome according to their stromal composition. However, the predictive power of aFib-RS remains superior to that of Fib-TBRS in stromal (CMS4) CRC tumors, which suggests that expression of aFib-RS can further identify CRC with tumor promoting/protective stroma.

Point 7 - The focus on IL11 in Figure 4, which has a very weak R correlation value in the patient data should be explained more in the results, otherwise I would suggest moving this to the supplementary. When compared to the strength of the data in Figure 5, with high correlations and extreme differentials, alongside really eloquent and convincing TBRI data, the IL11 data seems underwhelming.

We acknowledge this reviewers' point on the weaker correlation IL11 has with the aFib-RS signature in figure 4a. Hence, we further developed the study of IL11 in this revised version of the manuscript. Similar to aFib-RS, high *IL11* mRNA expression was an independent predictor of poor prognosis in the two analyzed patients cohorts (figure 4f, supplementary figure 5c,d). Accordingly, *IL11* levels were particularly increased in the CMS4 tumors subtype (figure 4g) associated with worse clinical outcome¹⁸. Increased expression of *IL11* also robustly segregated relapsing patients in this subset (figure 4h, supplementary figure 5d), thus indicating its superior predictive value over CMS classification. Of note, IL11 is an activator of STAT3 phosphorylation⁴⁵. As detailed in point 3, we assessed oxaliplatin effect in an immunocompetent murine model of aggressive CRC described by Tauriello and colleagues¹. IHC analyses showed increased P-STAT3 nuclear staining after therapy (supplementary figure 5a), indicating the activation upon treatment of IL11/STAT3 signaling previously associated with cancer cells aggressiveness³. Main text was modified accordingly.

Useful if possible to include:

- A) Can the authors show if oxaliplatin is taken up in normal stroma, perhaps in patient sample where this is a region of tissue separated from the tumour?
- B) Figure 2E: Can this be presented in the same way as figure 2b/c?
- C) The analyses in Figure 2 would benefit from a plot for aFIB-RS according to CMS (similar to Figure 6).
- D) While the patients samples will likely have been exposed to 5FU+oxali therapies, can the authors include some of the analyses on their fibroblast line with 5FU+oxali, alongside oxali alone, as it is unlikely that oxaliplatin would even be given as a mono-therapy for CRC. Typo: Line 166: do the authors mean Fig 3D?

We have now included the analysis of aFIB-RS according to CMS (figure 2e). This signature is particularly enriched in CMS4.

Current standard-of-care involves either fluoropyrimidines (5-fluorouracil, capecitabine) in early stage CRC or their combination with oxaliplatin in later stages. However, as demonstrated by the results of the MOSAIC trial¹⁰ and the works referenced in the introduction^{11,12}, oxaliplatin administration offers little benefit to fluoropyrimidines, and a high proportion of patients will still relapse. Yet, the addition of oxaliplatin to fluoropyrimidines regimen remains the main treatment option in aggressive CRC^{10,13,14}. Strengthening our concern, we saw that a major proportion of administered platinum, as seen in figure 1, was retained in the tumor stroma. Given the scope of the present manuscript, we used oxaliplatin alone as a system to understand the influence this drug exerts during tumor relapses. While we would not discourage the administration of oxaliplatin to CRC patients, the relevance of our results points to the clinical use of the genetic signatures from platinum-induced CAFs to predict the actual benefit of oxaliplatin. In addition, our data suggest that POSTN could be used as a biomarker for clinical diagnosis using formalin-fixed paraffin-embedded tumor samples to assist clinical decision-making. Main text was modified accordingly.

We thank this reviewer for spotting typo in line 166 that was corrected in the revised manuscript.

Reviewer #1 (Additional Remarks to the Author)

In their revised manuscript Linares et al addressed most of the concerns that I have raised. In particular, they nicely addressed the question of potential contribution of other cell types to protection from platinum (points 2,3). They also added analyses of the gene signature they found (to address comment 4) and the reasoning for investigating platinum alone (point 6) is convincing. Their response to comments 1 and 5 is not full.

They added a very valuable immunocompetent, orthotopic mouse model to address comment 1 which is important, and they stained for one more CAF marker (FAP) but it is not clear to me why they did not perform co-staining of CAF markers and the drug on the same slides.

We agree with this reviewer that the experimental setting description of drug uptake measurement deserves to be further clarified to address point 1. During LA-ICP-MS, the laser beam is focused on the sample surface to generate fine particles. The ablated particles are then transported to the secondary excitation source of the ICP-MS instrument for digestion and ionization of the sampled mass. The excited ions in the plasma torch are subsequently introduced to a mass spectrometer detector for elemental analysis. Hence, LA-ICP-MS results in the destruction of the tissue and prevents further analysis of the same area. Moreover, LA-ICP-MS resolution does not allow sub-cellular detection of platinum uptake in our experimental setting. Expert pathologist evaluation of cell morphology and serial section staining with FAP antibody were performed to overcome these limitations and results were displayed as follows in the main text.

“Histological analysis by expert pathologists and LA-ICP-MS bioimaging on tumor sections stained with hematoxylin and eosin revealed that oxaliplatin was predominantly retained in tumor areas enriched with fibroblasts. Accordingly, IHC analysis indicated that FAP expression –a marker of CAFs- overlapped with tumor areas displaying increased platinum uptake.”

In addition to above mentioned observations in patients, our *in vitro* and *in vivo* data indicate that fibroblasts -the most prominent cell type within the tumor microenvironment^{46,47}- display great resistance to oxaliplatin cytotoxicity and are particularly prone to absorb and retain platinum-based drug. Conversely, other stromal cells –endothelial and immune cells- are highly sensitive to the cytotoxic effect of chemotherapy and are cleared from the tumor upon treatment.

Taken together and as mentioned in the manuscript, our findings indicate that long term platinum retention during chemotherapy occurs in the TME of CRC patients and suggest that this process is largely dependent on resilient CAFs present at the time of treatment.

Similarly, they should have done costaining of POSTN with other markers rather than serial section staining. If this is technically too challenging they should provide an explanation. Multiplexed imaging and image analysis nowadays are widely available, and performing IHC (instead of IF) and performing manual assessment rather than automated image analysis is curious. If this is not feasible they should at least provide high magnification images that will clearly show that the drug

is inside fibroblasts and not just in stromal areas which are almost always a mix of CAFs with other cell types. This is important to support the claim that the CAFs actually accumulate the drug. Alternatively, the authors can say that CAFs mediate resistance, and tone down the claim on accumulation of drug.

We thank this reviewer for further developing his original comment regarding POSTN expression by CAFs (point 5) and agree that additional explanations are needed. POSTN expression by CRC CAFs has been already established in previous studies showing POSTN co-localization with alpha-SMA and vimentin –two markers of CAFs- by co-immunofluorescence staining as well as POSTN expression in CAFs by transcriptomics and by in situ hybridization^{34,42,48}. In the present study, we sought to complement these previous reports with information about FAP, another well-established marker of CAFs. Anti-FAP antibody is not adapted to immunofluorescence detection. However, immunohistochemistry and evaluation by expert pathologist are gold standards for biomarker characterization -expression pattern, localization and application- in the clinical setting-⁴⁹⁻⁵¹. Accordingly, we performed POSTN and FAP immunohistochemistry in consecutive sections of CRC tumors. Analysis by expert pathologist indicated that POSTN and FAP were expressed by tumor-associated fibroblasts, thus corroborating the POSTN expression by CRC CAFs reported in previous studies.

1. Tauriello DVF, Palomo-Ponce S, Stork D, Berenguer-Llergo A, Badia-Ramentol J, Iglesias M, et al. TGF β drives immune evasion in genetically reconstituted colon cancer metastasis. *Nature*. February 22, 2018;554(7693):538–543.
2. Cañellas-Socias A, Cortina C, Hernando-Momblona X, Palomo-Ponce S, Mulholland EJ, Turon G, et al. Metastatic recurrence in colorectal cancer arises from residual EMP1+ cells. *Nat* 2022 6117936. November 9, 2022 [cited November 17, 2022];611(7936):603–613. Available at: <https://www.nature.com/articles/s41586-022-05402-9>
3. Calon A, Espinet E, Palomo-Ponce S, Tauriello D V, Iglesias M, Cespedes M V, et al. Dependency of colorectal cancer on a TGF-beta-driven program in stromal cells for metastasis initiation. *Cancer Cell*. 2012/11/17. 2012;22(5):571–584. Available at: <http://www.ncbi.nlm.nih.gov/pubmed/23153532>
4. Coppé JP, Desprez PY, Krtolica A, Campisi J. The Senescence-Associated Secretory Phenotype: The Dark Side of Tumor Suppression. *Annu Rev Pathol*. February 2, 2010 [cited November 17, 2022];5:99. Available at: </pmc/articles/PMC4166495/>
5. Guo Y, Ayers JL, Carter KT, Wang T, Maden SK, Edmond D, et al. Senescence-associated tissue microenvironment promotes colon cancer formation through the secretory factor GDF15. *Aging Cell*. December 1, 2019 [cited April 10, 2021];18(6). Available at: </pmc/articles/PMC6826139/>
6. Hoare M, Ito Y, Kang TW, Weekes MP, Matheson NJ, Patten DA, et al. NOTCH1 mediates a switch between two distinct secretomes during senescence. *Nat Cell Biol*. September 1, 2016 [cited November 21, 2022];18(9):979. Available at: </pmc/articles/PMC5008465/>
7. Lasry A, Ben-Neriah Y. Senescence-associated inflammatory responses: Aging and cancer perspectives [Internet]. Vol. 36, Trends in Immunology. Elsevier Ltd 2015 [cited March 2,

2021]; 217–228. Available at:
<http://www.cell.com.sire.ub.edu/article/S147149061500040X/fulltext>

8. Hubackova S, Krejcikova K, Bartek J, Hodny Z. IL1- and TGFβ-Nox4 signaling, oxidative stress and DNA damage response are shared features of replicative, oncogene-induced, and drug-induced paracrine “bystander senescence.” *Aging (Albany NY)*. 2012 [cited November 21, 2022];4(12):932–951. Available at: <https://pubmed.ncbi.nlm.nih.gov/23385065/>
9. Meyerholz DK, Beck AP. Principles and approaches for reproducible scoring of tissue stains in research. *Lab Invest* 2018 987. May 30, 2018 [cited March 6, 2022];98(7):844–855. Available at: <https://www.nature.com/articles/s41374-018-0057-0>
10. André T, Boni C, Mounedji-Boudiaf L, Navarro M, Tabernero J, Hickish T, et al. Oxaliplatin, Fluorouracil, and Leucovorin as Adjuvant Treatment for Colon Cancer. *N Engl J Med*. June 3, 2004 [cited February 11, 2017];350(23):2343–2351. Available at: <http://www.nejm.org/doi/abs/10.1056/NEJMoa032709>
11. Rottenberg S, Disler C, Perego P. The rediscovery of platinum-based cancer therapy. *Nat Rev Cancer* 2020 211. October 30, 2020 [cited November 5, 2021];21(1):37–50. Available at: <https://www.nature.com/articles/s41568-020-00308-y>
12. Thiabaud G, He G, Sen S, Shelton KA, Baze WB, Segura L, et al. Oxaliplatin Pt(IV) prodrugs conjugated to gadolinium-texaphyrin as potential antitumor agents. *Proc Natl Acad Sci*. March 31, 2020 [cited November 5, 2021];117(13):7021–7029. Available at: <https://www.pnas.org/content/117/13/7021>
13. André T, Boni C, Navarro M, Tabernero J, Hickish T, Topham C, et al. Improved overall survival with oxaliplatin, fluorouracil, and leucovorin as adjuvant treatment in stage II or III colon cancer in the MOSAIC trial. *J Clin Oncol*. July 1, 2009;27(19):3109–3116.
14. Haller DG, Tabernero J, Maroun J, De Braud F, Price T, Van Cutsem E, et al. Capecitabine plus oxaliplatin compared with fluorouracil and folinic acid as adjuvant therapy for stage III colon cancer. *J Clin Oncol*. April 10, 2011 [cited May 6, 2021];29(11):1465–1471. Available at: <https://pubmed.ncbi.nlm.nih.gov/21383294/>
15. Sadanandam A, Lyssiotis CA, Homicsko K, Collisson EA, Gibb WJ, Wullschleger S, et al. A colorectal cancer classification system that associates cellular phenotype and responses to therapy. *Nat Med*. May 2013 [cited May 23, 2016];19(5):619–25. Available at: <http://www.ncbi.nlm.nih.gov/pubmed/23584089>
16. De Sousa E Melo F, Wang X, Jansen M, Fessler E, Trinh A, de Rooij LPMH, et al. Poor-prognosis colon cancer is defined by a molecularly distinct subtype and develops from serrated precursor lesions. *Nat Med*. May 2013 [cited May 23, 2016];19(5):614–8. Available at: <http://www.ncbi.nlm.nih.gov/pubmed/23584090>
17. Roepman P, Schlicker A, Tabernero J, Majewski I, Tian S, Moreno V, et al. Colorectal cancer intrinsic subtypes predict chemotherapy benefit, deficient mismatch repair and epithelial-to-mesenchymal transition. *Int J Cancer*. February 1, 2014 [cited January 24, 2021];134(3):552–562. Available at: <https://pubmed.ncbi.nlm.nih.gov/23852808/>
18. Guinney J, Dienstmann R, Wang X, de Reyniès A, Schlicker A, Sonesson C, et al. The consensus molecular subtypes of colorectal cancer. *Nat Med*. November 2015 [cited May

- 23, 2016];21(11):1350–6. Available at: <http://www.ncbi.nlm.nih.gov/pubmed/26457759>
19. Li C, Lau HCH, Zhang X, Yu J. Mouse Models for Application in Colorectal Cancer: Understanding the Pathogenesis and Relevance to the Human Condition. *Biomed 2022, Vol 10, Page 1710*. July 15, 2022 [cited November 17, 2022];10(7):1710. Available at: <https://www.mdpi.com/2227-9059/10/7/1710/htm>
 20. Fábio GC. Surgical treatment of familial adenomatous polyposis: dilemmas and current recommendations. *World J Gastroenterol*. November 28, 2014 [cited November 17, 2022];20(44):16620–16629. Available at: <https://pubmed.ncbi.nlm.nih.gov/25469031/>
 21. Calon A, Espinet E, Palomo-Ponce S, Tauriello DVF, Iglesias M, Céspedes MV, et al. Dependency of Colorectal Cancer on a TGF- β -Driven Program in Stromal Cells for Metastasis Initiation. *Cancer Cell*. November 13, 2012 [cited June 11, 2018];22(5):571–584. Available at: <http://www.ncbi.nlm.nih.gov/pubmed/23153532>
 22. Pickup M, Novitskiy S, Moses HL. The roles of TGF β in the tumour microenvironment. *Nat Rev Cancer*. November 2013 [cited January 29, 2015];13(11):788–99. Available at: <http://www.pubmedcentral.nih.gov/articlerender.fcgi?artid=4025940&tool=pmcentrez&rendertype=abstract>
 23. Massagué J. TGF β in Cancer [Internet]. Vol. 134, Cell. NIH Public Access 2008 [cited December 3, 2020]; 215–230. Available at: </pmc/articles/PMC3512574/?report=abstract>
 24. Chandra R, Karalis JD, Liu C, Murimwa GZ, Park JV, Heid CA, et al. The Colorectal Cancer Tumor Microenvironment and Its Impact on Liver and Lung Metastasis. *Cancers 2021, Vol 13, Page 6206*. December 9, 2021 [cited October 22, 2022];13(24):6206. Available at: <https://www.mdpi.com/2072-6694/13/24/6206/htm>
 25. Calon A, Tauriello DVF, Batlle E. TGF-beta in CAF-mediated tumor growth and metastasis. *Semin Cancer Biol*. April 2014 [cited May 24, 2016];25:15–22. Available at: <http://www.ncbi.nlm.nih.gov/pubmed/24412104>
 26. Dörr JR, Yu Y, Milanovic M, Beuster G, Zasada C, Däbritz JHM, et al. Synthetic lethal metabolic targeting of cellular senescence in cancer therapy. *Nature*. 2013 [cited October 22, 2022];501(7467):421–425. Available at: <https://pubmed.ncbi.nlm.nih.gov/23945590/>
 27. Lopes-Paciencia S, Saint-Germain E, Rowell MC, Ruiz AF, Kalegari P, Ferbeyre G. The senescence-associated secretory phenotype and its regulation. *Cytokine*. May 1, 2019 [cited October 22, 2022];117:15–22. Available at: <https://pubmed.ncbi.nlm.nih.gov/30776684/>
 28. Ou HL, Hoffmann R, González-López C, Doherty GJ, Korkola JE, Muñoz-Espín D. Cellular senescence in cancer: from mechanisms to detection. *Mol Oncol*. October 1, 2021 [cited October 22, 2022];15(10):2634. Available at: </pmc/articles/PMC8486596/>
 29. Coryell PR, Diekman BO, Loeser RF. Mechanisms and therapeutic implications of cellular senescence in osteoarthritis. *Nat Rev Rheumatol*. January 1, 2021 [cited October 22, 2022];17(1):47. Available at: </pmc/articles/PMC8035495/>
 30. Kalluri R. The biology and function of fibroblasts in cancer [Internet]. Vol. 16, Nature Reviews Cancer. Nature Publishing Group 2016 [cited November 30, 2020]; 582–598. Available at: <https://pubmed.ncbi.nlm.nih.gov/27550820/>

31. Chen H, Chen H, Liang J, Gu X, Zhou J, Xie C, et al. TGF- β 1/IL-11/MEK/ERK signaling mediates senescence-associated pulmonary fibrosis in a stress-induced premature senescence model of Bmi-1 deficiency. *Exp Mol Med*. January 1, 2020 [cited August 2, 2021];52(1):130–151. Available at: [/pmc/articles/PMC7000795/](https://pubmed.ncbi.nlm.nih.gov/33317907/)
32. Shin SY, Choi C, Lee HG, Lim Y, Lee YH. Transcriptional regulation of the interleukin-11 gene by oncogenic Ras. *Carcinogenesis*. December 1, 2012 [cited October 22, 2022];33(12):2467–2476. Available at: <https://academic.oup.com/carcin/article/33/12/2467/2464281>
33. Kongkavitoon P, Butta P, Sanpavat A, Bhattarakosol P, Tangtanatakul P, Wongprom B, et al. Regulation of periostin expression by Notch signaling in hepatocytes and liver cancer cell lines. *Biochem Biophys Res Commun*. November 30, 2018 [cited October 22, 2022];506(3):739–745. Available at: <https://pubmed.ncbi.nlm.nih.gov/30384995/>
34. Ma H, Wang J, Zhao X, Wu T, Huang Z, Chen D, et al. Periostin Promotes Colorectal Tumorigenesis through Integrin-FAK-Src Pathway-Mediated YAP/TAZ Activation. *Cell Rep*. January 21, 2020;30(3):793-806.e6.
35. Yashiro R, Nagasawa T, Kiji M, Hormdee D, Kobayashi H, Koshy G, et al. Transforming growth factor-beta stimulates interleukin-11 production by human periodontal ligament and gingival fibroblasts. *J Clin Periodontol*. March 2006 [cited October 22, 2022];33(3):165–171. Available at: <https://pubmed.ncbi.nlm.nih.gov/16489941/>
36. Tang W, Yang L, Yang YC, Leng SX, Elias JA. Transforming growth factor-beta stimulates interleukin-11 transcription via complex activating protein-1-dependent pathways. *J Biol Chem*. March 6, 1998 [cited October 22, 2022];273(10):5506–5513. Available at: <https://pubmed.ncbi.nlm.nih.gov/9488674/>
37. Nanri Y, Nunomura S, Terasaki Y, Yoshihara T, Hirano Y, Yokozaki Y, et al. The cross-talk between TGF- β and periostin can be targeted for pulmonary fibrosis. In European Respiratory Society (ERS) 2019; PA1292.
38. Ouanouki A, Lamy S, Annabi B. Periostin, a signal transduction intermediate in TGF- β -induced EMT in U-87MG human glioblastoma cells, and its inhibition by anthocyanidins. *Oncotarget*. April 4, 2018 [cited October 22, 2022];9(31):22023. Available at: [/pmc/articles/PMC5955165/](https://pubmed.ncbi.nlm.nih.gov/30384995/)
39. Yue H, Li W, Chen R, Wang J, Lu X, Li J. Stromal POSTN induced by TGF- β 1 facilitates the migration and invasion of ovarian cancer. *Gynecol Oncol*. February 1, 2021 [cited October 22, 2022];160(2):530–538. Available at: <https://pubmed.ncbi.nlm.nih.gov/33317907/>
40. Corden B, Adami E, Sweeney M, Schafer S, Cook SA. IL-11 in cardiac and renal fibrosis: Late to the party but a central player. *Br J Pharmacol*. April 1, 2020 [cited October 22, 2022];177(8):1695–1708. Available at: <https://onlinelibrary.wiley.com/doi/full/10.1111/bph.15013>
41. Schafer S, Viswanathan S, Widjaja AA, Lim WW, Moreno-Moral A, DeLaughter DM, et al. IL-11 is a crucial determinant of cardiovascular fibrosis. *Nature*. 2017 [cited October 22, 2022];552(7683):110–115. Available at: <https://pubmed.ncbi.nlm.nih.gov/29160304/>
42. Calon A, Lonardo E, Berenguer-Llergo A, Espinet E, Hernando-momblona X, Iglesias M, et al.

Stromal gene expression defines poor-prognosis subtypes in colorectal cancer. *Nat Genet.* 2015 [cited May 24, 2016];47(February):320–329. Available at: <http://dx.doi.org/10.1038/ng.3225>

43. Dunne PD, McArt DG, Bradley CA, O'Reilly PG, Barrett HL, Cummins R, et al. Challenging the cancer molecular stratification dogma: Intratumoral heterogeneity undermines consensus molecular subtypes and potential diagnostic value in colorectal cancer. *Clin Cancer Res.* May 5, 2016 [cited May 23, 2016]; Available at: <http://www.ncbi.nlm.nih.gov/pubmed/27151745>
44. Michiels S, Ternès N, Rotolo F. Statistical controversies in clinical research: prognostic gene signatures are not (yet) useful in clinical practice. *Ann Oncol.* December 1, 2016 [cited August 31, 2022];27(12):2160–2167. Available at: <http://www.annalsofncology.org/article/S0923753419365317/fulltext>
45. Ernst M, Putoczki TL. Molecular pathways: IL11 as a tumor-promoting cytokine-translational implications for cancers. *Clin Cancer Res.* November 15, 2014 [cited November 19, 2022];20(22):5579–5588. Available at: <https://aacrjournals.org/clincancerres/article/20/22/5579/117468/Molecular-Pathways-IL11-as-a-Tumor-Promoting>
46. Pietras K, Ostman A. Hallmarks of cancer: interactions with the tumor stroma. *Exp Cell Res.* 2010/03/10. 2010;316(8):1324–1331. Available at: <http://www.ncbi.nlm.nih.gov/pubmed/20211171>
47. Wang W, Cheng B, Yu Q. Cancer-associated fibroblasts as accomplices to confer therapeutic resistance in cancer. *Cancer Drug Resist.* September 7, 2022 [cited December 24, 2022];5(4):889–901. Available at: <https://cdrjournal.com/article/view/5147>
48. Kikuchi Y, Kashima TG, Nishiyama T, Shimazu K, Morishita Y, Shimazaki M, et al. Periostin is expressed in pericryptal fibroblasts and cancer-associated fibroblasts in the colon. *J Histochem Cytochem.* August 2008 [cited December 24, 2022];56(8):753–764. Available at: <https://pubmed.ncbi.nlm.nih.gov/18443362/>
49. Van Cutsem E, Cervantes A, Adam R, Sobrero A, Van Krieken JH, Aderka D, et al. ESMO consensus guidelines for the management of patients with metastatic colorectal cancer. *Ann Oncol.* August 1, 2016 [cited May 3, 2021];27(8):1386–1422. Available at: <https://pubmed.ncbi.nlm.nih.gov/27380959/>
50. Schmoll HJ, Van cutsem E, Stein A, Valentini V, Glimelius B, Haustermans K, et al. ESMO Consensus Guidelines for management of patients with colon and rectal cancer. A personalized approach to clinical decision making. *Ann Oncol.* October 1, 2012 [cited December 24, 2022];23(10):2479–2516. Available at: <http://www.annalsofncology.org/article/S0923753419379803/fulltext>
51. Cervantes A, Adam R, Roselló S, Arnold D, Normanno N, Taïeb J, et al. Metastatic colorectal cancer: ESMO Clinical Practice Guideline for diagnosis, treatment and follow-up. *Ann Oncol Off J Eur Soc Med Oncol.* October 2022;

REVIEWERS' COMMENTS

Reviewer #1 (Remarks to the Author):

In their revised manuscript Linares et al addressed most of the concerns that I have raised. In particular, they nicely addressed the question of potential contribution of other cell types to protection from platinum (points 2,3). They also added analyses of the gene signature they found (to address comment 4) and the reasoning for investigating platinum alone (point 6) is convincing. Their response to comments 1 and 5 is not full. They added a very valuable immunocompetent, orthotopic mouse model to address comment 1 which is important, and they stained for one more CAF marker (FAP) but it is not clear to me why they did not perform co-staining of CAF markers and the drug on the same slides. Similarly, they should have done costaining of POSTN with other markers rather than serial section staining. If this is technically too challenging they should provide an explanation. Multiplexed imaging and image analysis nowadays are widely available, and performing IHC (instead of IF) and performing manual assessment rather than automated image analysis is curious. If this is not feasible they should at least provide high magnification images that will clearly show that the drug is inside fibroblasts and not just in stromal areas which are almost always a mix of CAFs with other cell types. This is important to support the claim that the CAFs actually accumulate the drug. Alternatively, the authors can say that CAFs mediate resistance, and tone down the claim on accumulation of drug.

Reviewer #2 (Remarks to the Author):

The authors have addressed my concerns. No further questions.

Reviewer #4 (Remarks to the Author):

The revised article has address all my initial comments. I commend the authors for the substantial new data presented in this revised version, which I feel have strengthened the conclusions.

A very eloquent study, and I fully support its acceptance.

Response to Reviewer #1's additional remarks to the authors

In their revised manuscript Linares et al addressed most of the concerns that I have raised. In particular, they nicely addressed the question of potential contribution of other cell types to protection from platinum (points 2,3). They also added analyses of the gene signature they found (to address comment 4) and the reasoning for investigating platinum alone (point 6) is convincing. Their response to comments 1 and 5 is not full.

They added a very valuable immunocompetent, orthotopic mouse model to address comment 1 which is important, and they stained for one more CAF marker (FAP) but it is not clear to me why they did not perform co-staining of CAF markers and the drug on the same slides.

We agree with this reviewer that the experimental setting description of drug uptake measurement deserves to be further clarified to address point 1. During LA-ICP-MS, the laser beam is focused on the sample surface to generate fine particles. The ablated particles are then transported to the secondary excitation source of the ICP-MS instrument for digestion and ionization of the sampled mass. The excited ions in the plasma torch are subsequently introduced to a mass spectrometer detector for elemental analysis. Hence, LA-ICP-MS results in the destruction of the tissue and prevents further analysis of the same area. Moreover, LA-ICP-MS resolution does not allow sub-cellular detection of platinum uptake in our experimental setting. Expert pathologist evaluation of cell morphology and serial section staining with FAP antibody were performed to overcome these limitations and results were displayed as follows in the main text.

“Histological analysis by expert pathologists and LA-ICP-MS bioimaging on tumor sections stained with hematoxylin and eosin revealed that oxaliplatin was predominantly retained in tumor areas enriched with fibroblasts. Accordingly, IHC analysis indicated that FAP expression –a marker of CAFs- overlapped with tumor areas displaying increased platinum uptake.”

In addition to above mentioned observations in patients, our *in vitro* and *in vivo* data indicate that fibroblasts -the most prominent cell type within the tumor microenvironment^{1,2}- display great resistance to oxaliplatin cytotoxicity and are particularly prone to absorb and retain platinum-based drug. Conversely, other stromal cells –endothelial and immune cells- are highly sensitive to the cytotoxic effect of chemotherapy and are cleared from the tumor upon treatment.

Taken together and as mentioned in the manuscript, our findings indicate that long term platinum retention during chemotherapy occurs in the TME of CRC patients and suggest that this process is largely dependent on resilient CAFs present at the time of treatment.

Similarly, they should have done costaining of POSTN with other markers rather than serial section staining. If this is technically too challenging they should provide an explanation. Multiplexed imaging and image analysis nowadays are widely available, and performing IHC (instead of IF) and performing manual assessment rather than automated image analysis is curios. If this is not feasible they should at least provide high magnification images that will clearly show that the drug

is inside fibroblasts and not just in stromal areas which are almost always a mix of CAFs with other cell types. This is important to support the claim that the CAFs actually accumulate the drug. Alternatively, the authors can say that CAFs mediate resistance, and tone down the claim on accumulation of drug.

We thank this reviewer for further developing his original comment regarding POSTN expression by CAFs (point 5) and agree that additional explanations are needed. POSTN expression by CRC CAFs has been already established in previous studies showing POSTN co-localization with alpha-SMA and vimentin –two markers of CAFs- by co-immunofluorescence staining as well as POSTN expression in CAFs by transcriptomics and by in situ hybridization³⁻⁵. In the present study, we sought to complement these previous reports with information about FAP, another well-established marker of CAFs. Anti-FAP antibody is not adapted to immunofluorescence detection. However, immunohistochemistry and evaluation by expert pathologist are gold standards for biomarker characterization -expression pattern, localization and application- in the clinical setting-⁶⁻⁸. Accordingly, we performed POSTN and FAP immunohistochemistry in consecutive sections of CRC tumors. Analysis by expert pathologist indicated that POSTN and FAP were expressed by tumor-associated fibroblasts, thus corroborating the POSTN expression by CRC CAFs reported in previous studies.

1. Pietras K, Ostman A. Hallmarks of cancer: interactions with the tumor stroma. *Exp Cell Res.* 2010/03/10. 2010;316(8):1324–1331. Available at: <http://www.ncbi.nlm.nih.gov/pubmed/20211171>
2. Wang W, Cheng B, Yu Q. Cancer-associated fibroblasts as accomplices to confer therapeutic resistance in cancer. *Cancer Drug Resist.* September 7, 2022 [cited December 24, 2022];5(4):889–901. Available at: <https://cdrjournal.com/article/view/5147>
3. Calon A, Lonardo E, Berenguer-Llargo A, Espinet E, Hernando-momblona X, Iglesias M, et al. Stromal gene expression defines poor-prognosis subtypes in colorectal cancer. *Nat Genet.* 2015 [cited May 24, 2016];47(February):320–329. Available at: <http://dx.doi.org/10.1038/ng.3225>
4. Ma H, Wang J, Zhao X, Wu T, Huang Z, Chen D, et al. Periostin Promotes Colorectal Tumorigenesis through Integrin-FAK-Src Pathway-Mediated YAP/TAZ Activation. *Cell Rep.* January 21, 2020;30(3):793-806.e6.
5. Kikuchi Y, Kashima TG, Nishiyama T, Shimazu K, Morishita Y, Shimazaki M, et al. Periostin is expressed in pericryptal fibroblasts and cancer-associated fibroblasts in the colon. *J Histochem Cytochem.* August 2008 [cited December 24, 2022];56(8):753–764. Available at: <https://pubmed.ncbi.nlm.nih.gov/18443362/>
6. Van Cutsem E, Cervantes A, Adam R, Sobrero A, Van Krieken JH, Aderka D, et al. ESMO consensus guidelines for the management of patients with metastatic colorectal cancer. *Ann Oncol.* August 1, 2016 [cited May 3, 2021];27(8):1386–1422. Available at: <https://pubmed.ncbi.nlm.nih.gov/27380959/>
7. Schmoll HJ, Van cutsem E, Stein A, Valentini V, Glimelius B, Haustermans K, et al. ESMO Consensus Guidelines for management of patients with colon and rectal cancer. A

personalized approach to clinical decision making. *Ann Oncol*. October 1, 2012 [cited December 24, 2022];23(10):2479–2516. Available at: <http://www.annalsofoncology.org/article/S0923753419379803/fulltext>

8. Cervantes A, Adam R, Roselló S, Arnold D, Normanno N, Taïeb J, et al. Metastatic colorectal cancer: ESMO Clinical Practice Guideline for diagnosis, treatment and follow-up†. *Ann Oncol*. October 2022. (22)04192-8. Available at: [https://www.annalsofoncology.org/article/S0923-7534\(22\)04192-8/fulltext](https://www.annalsofoncology.org/article/S0923-7534(22)04192-8/fulltext)